# EHRSHOT: An EHR Benchmark for Few-Shot Evaluation of Foundation Models

**Michael Wornow**[*]
Department of Computer Science
Stanford University
mwornow@stanford.edu

**Rahul Thapa**[*]
Center for Biomedical Informatics Research
Stanford University
rthapa84@stanford.edu

**Ethan Steinberg**
Department of Computer Science
Stanford University
ethan@stanford.edu

**Jason A. Fries**[†]
Center for Biomedical Informatics Research
Stanford University
jfries@stanford.edu

**Nigam H. Shah**[†]
Center for Biomedical Informatics Research
Clinical Excellence Research Center
Stanford University
Technology and Digital Solutions
Stanford Healthcare
nigam@stanford.edu

## Abstract

While the general machine learning (ML) community has benefited from public datasets, tasks, and models, the progress of ML in healthcare has been hampered by a lack of such shared assets. The success of foundation models creates new challenges for healthcare ML by requiring access to shared pretrained models to validate performance benefits. We help address these challenges through three contributions. First, we publish a new dataset, EHRSHOT, which contains de-identified structured data from the electronic health records (EHRs) of 6,739 patients from Stanford Medicine. Unlike MIMIC-III/IV and other popular EHR datasets, EHRSHOT is longitudinal and not restricted to ICU/ED patients. Second, we publish the weights of CLMBR-T-base, a 141M parameter clinical foundation model pretrained on the structured EHR data of 2.57M patients. We are one of the first to fully release such a model for coded EHR data; in contrast, most prior models released for clinical data (e.g. GatorTron, ClinicalBERT) only work with unstructured text and cannot process the rich, structured data within an EHR. We provide an end-to-end pipeline for the community to validate and build upon its performance. Third, we define 15 few-shot clinical prediction tasks, enabling evaluation of foundation models on benefits such as sample efficiency and task adaptation. Our model and dataset are available via a research data use agreement from our website. Code to reproduce our results is available here.

---

[*]Equal contribution
[†]Equal senior authorship

37th Conference on Neural Information Processing Systems (NeurIPS 2023) Track on Datasets and Benchmarks.

# 1 Introduction

Open datasets, code, and models have been essential in advancing machine learning (ML) over the past decade [33, 45, 18]. Though the benefits of open code and data are well known [39, 26], there is currently a dearth of publicly available datasets and pretrained models for electronic health records (EHRs), which makes conducting reproducible research challenging [37, 22].

This is especially problematic in the era of foundation models (FMs), which hold tremendous promise for clinical applications [23]. The ability of a shared FM to generalize across health systems would be highly valuable, as most hospitals lack the computational resources to train such models [35]. Yet many of the purported benefits of clinical FMs, such as sample efficiency and task adaptability, remain difficult to evaluate due to reproducibility and data access issues [37].

Unfortunately, most existing EHR datasets (e.g., MIMIC-III/IV [16, 15], eICU [27], AmsterdamUM-Cdb [44], and HiRID [6]) narrowly focus on the intensive care unit (ICU), which provides a limited snapshot of a patient's overall health trajectory and limits what tasks can be evaluated [46]. Access to a patient's complete medical timeline, referred to as "longitudinal" data, offers a more realistic representation of the breadth of information available to a health system. Longitudinal EHR data, however, remains scarce. The few public datasets that exist, such as the CPRD [11] and UK BioBank [2], lack consensus on shared evaluation tasks / data processing pipelines and require navigating a research protocol review process, which creates challenges when curating shared ML workflows [43].

While the limitations of prior benchmarks were less apparent when developing small-scale, task-specific models, their utility is limited for evaluating FMs on task adaptation, few-shot learning, and other properties of large-scale, self-supervised models [1, 30]. Clinical FMs surface new questions, and a dataset for evaluating such FMs should contain a diverse range of tasks in low-label settings with longitudinal data [21]. Most importantly, such a benchmark should also release the weights of its pretrained models so the community can reproduce and build upon its results. Unfortunately, few FMs trained on EHR data have had their model weights published [48].

Our work helps address both shortcomings – a lack of public EHR datasets and pretrained clinical FMs – as one of the first combined releases of a research dataset and FM trained on EHR data. We outline our three primary contributions towards more reproducible ML for healthcare below:

1. We release EHRSHOT, a longitudinal EHR benchmark for the few-shot evaluation of clinical FMs. EHRSHOT contains the **full coded medical timelines of 6,739 patients** from Stanford Medicine. Records include demographics, diagnoses, procedures, laboratory results, medications, and other structured data, for a total of 41.6 million clinical events across 921,499 encounters. EHRSHOT contains an average of **2.3x more clinical events and 95.2x more encounters per patient than MIMIC-IV** [15] and, unlike the majority of existing benchmarks, includes patients not seen in the ICU or emergency department (ED).

2. We publish the weights of a **141M parameter transformer-based foundation model** (CLMBR-T-base) pretrained on the deidentified structured data of **2.57M patients' EHRs**. CLMBR-T-base was trained in a self-supervised manner to autoregressively predict the next code in a patient's timeline given their previous codes [41]. We are among the first to publish the full weights of such a clinical FM [48] for the community to evaluate and build upon. Researchers who leverage our model can benefit from both improved downstream task accuracy and cost savings by shortcutting the model development process.

3. We define a new few-shot benchmark of **15 patient classification tasks.** Several tasks have naturally low prevalence, creating a realistic setting for few-shot experimentation. While our pretrained model offers significant AUROC/AUPRC gains in few-shot settings over a traditional supervised baseline, we demonstrate that there remains significant room for improvement on many of our tasks.

Our overall workflow is shown in Figure 1. We publish the full code to replicate our results here: https://github.com/som-shahlab/ehrshot-benchmark. We also publish the full weights of our pretrained clinical foundation model, as well as the EHRSHOT dataset and task labels, under a non-commercial data usage agreement here: https://ehrshot.stanford.edu.

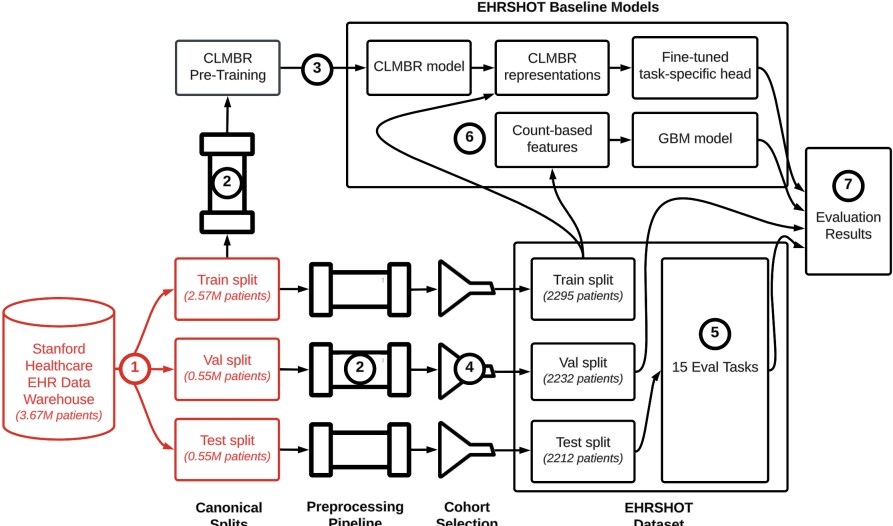

Figure 1: Overview of EHRSHOT. Black boxes represent open source code, data, and model weights. Red boxes are private data. (1) Starting with a source EHR database of 3.67M patients, we define a global train/val/test split across all patients. (2) We use an open source EHR preprocessing package called FEMR to transform our data. We keep all structured data (diagnoses, medications, labs, etc.) and discard images and clinical text. (3) We use the 2.57M patients in our global train split to pre-train a foundation model, CLMBR-T-base [41] (4) We filter the source database down to a cohort of 6,739 patients, which we use for EHRSHOT. (5) We define 15 few-shot classification tasks and label each patient accordingly. (6) We test two baseline models for each task: our pretrained CLMBR-T-base and a count-based GBM model [31]. (7) We measure the AUROC and AUPRC of each model on each task, and share the results in Section 5.

## 2  Related Work

One of the most popular EHR datasets made accessible to researchers is MIMIC-III, which contains roughly 40,000 patients seen in the intensive care unit (ICU) of Beth Israel Deaconess Medical Center in Boston, Massachusetts, between 2001 and 2012 [16]. Other public datasets include eICU [27], HiRID [6], AmsterdamUMCdb [44], CPRD [11], MIMIC-IV [15], and the UK BioBank [2].

Most of the aforementioned datasets are narrowly scoped to a single department: the ICU [16, 27, 6, 44]. This makes it impossible to capture a patient's full health trajectory to the extent that an academic medical center or health system would know of the patients it treats. Other datasets such as MIMIC-IV include data from multiple departments, but are still heavily anchored to the ICU, as only patients admitted for an ICU/ED visit are included [15]. In contrast, our work releases the full longitudinal EHR of patients across all departments of a major academic medical center, thus providing a more realistic setting for general prediction making.

Prior work has also typically relied on the creation of bespoke schemas to store their data. These custom schemas greatly increase the difficulty of transferring models across datasets and sites [43]. In contrast, the data preprocessing pipeline that we use is capable of ingesting both EHRSHOT as well as any dataset that follows the Observational Medical Outcomes Partnership Common Data Model (OMOP-CDM), an open community data standard for sharing EHRs used by over 100 health systems [34]. More details on our data preprocessing pipeline can be found in the Appendix in Section C.5.

Previously published EHR datasets typically only provide raw data. Thus, significant additional effort has been devoted to building standardized preprocessing pipelines, patient splits, and task definitions on top of these datasets [10, 29, 22]. These add-on benchmarks, however, are still limited by the narrow scope of their underlying data, and many recycle the same core set of tasks (e.g. in-patient mortality, long length-of-stay, ICU transfer, and ICD code prediction) [29, 10, 9]. Additionally, these benchmarks are typically not created with the purpose of measuring a pretrained model's few-shot

performance [21]. This limits their utility in assessing the key value propositions of foundation models, such as improved sample efficiency and adaptation to diverse tasks.

On the modeling side, substantial literature exists on training FMs for EHR data [28, 19, 32, 41, 24]. However, the vast majority of these FMs have never had their weights published [48]. This greatly hinders reproducibility and makes cross-model evaluations difficult. Worse, this lack of sharing undermines a primary advantage of FMs: transfer learning, i.e. the ability to use the pretrained weights of an existing FM to shortcut model development for other tasks [1].

EHRSHOT aims to fill several of these gaps by providing a longitudinal EHR benchmark specifically geared towards few-shot evaluation of pretrained FMs. EHRSHOT is built on top of a cross-site interoperable standard (OMOP-CDM), and leverages an open source data preprocessing pipeline to allow other researchers to reproduce our results end-to-end. Additionally, we release the weights of the clinical foundation model that we pretrain and evaluate, one of the first to do so. We provide additional points of comparison in Table 1.

Table 1: Comparison of our work to existing EHR benchmarks. Checkmark indicates full support, asterisk represents properties that are semi-supported.

| Benchmark | Source Dataset | EHR Properties | | | Evaluation | | Reproducibility | | |
| | | ICU/ED Visits | Other Visits | # of Patients | # of Tasks | Few Shot | Dataset via DUA | Preprocessing Code | Model Weights |
|---|---|---|---|---|---|---|---|---|---|
| MIMIC-Extract [47] | MIMIC-III | ✓ | – | 34k | 5 | – | ✓ | ✓ | – |
| Purushotham 2018 [29] | MIMIC-III | ✓ | – | 35k | 3 | – | ✓ | ✓ | – |
| Harutyunyan 2019 [10] | MIMIC-III | ✓ | – | 33k | 4 | – | ✓ | ✓ | – |
| Gupta 2022 [9] | MIMIC-IV | ✓ | * | 257k | 4 | – | ✓ | ✓ | – |
| COP-E-CAT [20] | MIMIC-IV | ✓ | * | 257k | 4 | – | ✓ | ✓ | – |
| Xie 2022 [49] | MIMIC-IV | ✓ | * | 216k | 3 | – | ✓ | ✓ | – |
| eICU [36] | eICU | ✓ | – | 73k | 4 | – | ✓ | ✓ | – |
| EHR PT [21] | MIMIC-III / eICU | ✓ | – | 86k | 11 | ✓ | ✓ | ✓ | – |
| FIDDLE [43] | MIMIC-III / eICU | ✓ | – | 157k | 3 | – | ✓ | ✓ | – |
| HiRID-ICU [50] | HiRID | ✓ | – | 33k | 6 | – | ✓ | ✓ | – |
| Solares 2020 [38] | CPRD | ✓ | ✓ | 4M | 2 | – | – | – | – |
| **EHRSHOT** | Stanford Medicine | ✓ | ✓ | 7k | 15 | ✓ | ✓ | ✓ | ✓ |

## 3 Dataset

We are releasing EHRSHOT (pronounced "earshot"), an EHR benchmark for few-shot evaluation of foundation models. EHRSHOT is a collection of 6,739 unique patients with canonical train/validation/test splits and corresponding labels for 15 classification tasks. We also provide canonical $k$-shot samples for each few-shot evaluation task. Unlike prior EHR benchmarks focused on task-specific supervised models [21] for specific episodes of care, e.g. admission to the ICU [10, 27], our benchmark is designed for evaluating pretrained FMs on a broad range of tasks using the depth of information that a health system would typically possess for its patients. EHRSHOT is provided as a set of CSV files. It is essentially a lightweight serialization of the OMOP-CDM format. Please see Section C.4 in the Appendix for additional details on the dataset format.

EHRSHOT contains a total of 41.6 million coded observations (e.g. diagnoses, procedures, medications, lab results, etc.) and 921,499 unique visits across 6,739 patients. We exclude all patients less than 19 years of age or greater than 88 years of age. We also exclude patients with less than 10 total clinical events in their record. We include statistics of EHRSHOT's cohort demographics in Table 2 and Appendix Table 4, and histograms of patient characteristics in Appendix Figure 4.

### 3.1 Data Source

We sourced the data for our benchmark from the Stanford Medicine Research Data Repository (STARR) [5], which contains EHR data from both Stanford Health Care (primarily adult care) and Lucile Packard Children's Hospital (primarily pediatric care). The source dataset is structured according to the Observational Medical Outcomes Partnership Common Data Model (OMOP-CDM) [12] and comprises a total of 3.67M unique patients from 1990 to February 8th, 2023 [5]. Of these patients, 2.57M (70%) are used for training and 0.55M (15%) for validation of the foundation model

Table 2: Summary statistics on the number of events, visits, and length of patient timelines in EHRSHOT.

| Attribute | | Train | Val | Test | All Splits |
|---|---|---|---|---|---|
| **Number of Events** | Min | 10 | 10 | 10 | 10 |
| | Mean | 5942 | 6758 | 5826 | 6174 |
| | Max | 113466 | 199913 | 129704 | 199913 |
| **Number of Visits** | Min | 0 | 0 | 0 | 0 |
| | Mean | 127 | 147 | 134 | 136 |
| | Max | 2099 | 2397 | 2023 | 2397 |
| **Timeline Length (yrs)** | Min | 19 | 19 | 19 | 19 |
| | Mean | 59 | 59 | 58 | 59 |
| | Max | 88 | 88 | 88 | 88 |

that we release, CLMBR-T-base, the details of which we discuss in Section 4. All data that we work with is deidentified, and hence, our study did not require Institutional Review Board approval [5].

This source database contains demographics (e.g. age, sex, race), diagnoses, procedures, laboratory results, medication prescriptions, and other coded clinical observations, which we preserve. While the source database also contains clinical notes, we remove these in our released benchmark. We describe how we selected our patient cohort from this source dataset in the Appendix in Section C.6. We apply a few additional transformations on top of those described in [5] to prevent data leakage and fix timestamp issues, which are detailed in Section C.5 in the Appendix.

For our data preprocessing pipeline, we use the **Framework for Electronic Medical Records (FEMR)** library, which we developed in parallel to this work. FEMR is a Python library that supports the ingestion of multiple EHR data formats (e.g. OMOP, MIMIC, etc.) and provides a unified interface for building machine learning models on top of such data at scale. The full codebase is available on Github here: https://github.com/som-shahlab/femr/.

Additionally, all of the code used to generate the dataset for EHRSHOT can be found here: https://github.com/som-shahlab/ehrshot-benchmark.

### 3.2 Tasks

We define 15 tasks as part of our benchmark, as listed in Table 3. We selected these tasks based on clinician input as well as alignment with prior benchmarks [10, 8]. The tasks that we consider can be broadly grouped into the following 4 categories: (1) Operational Outcomes, (2) Anticipating Lab Test Values, (3) Assignment of New Diagnoses, (4) Anticipating Chest X-ray Findings.

All tasks are classification tasks. We include a total of nine binary classification tasks (*Operational Outcomes* and *Assignment of New Diagnoses*), five 5-way multiclass tasks (*Anticipating Lab Test Values*), and one 14-way multilabel task (*Anticipating Chest X-ray Findings*). The size of each task's subcohort, as well as the prevalence of positive labels, is detailed in Table 3. For example, there are 552 positive labels within the test cohort for the Long Length of Stay task, while there are 2,195 total labels, meaning there are 1,643 negative labels. As there are only 1,238 unique patients in this task's test cohort, some patients have multiple labels assigned to them.

In the Appendix, we define the precise prediction windows for each task in Table 7 and the definition of each task in Section C.3. We also provide a visualization of our 4 task categories in Figure 2.

## 4 Baseline Models

We measure the performance of two baseline models on our dataset: (1) a gradient boosting machine (GBM) that uses count-based featurizations of patients to make predictions, (2) an autoregressive language model ("CLMBR-T-base") that ingests medical codes as tokens and was pretrained on the full longitudinal structured EHRs of 2.57M patients from our source institution [41, 8].

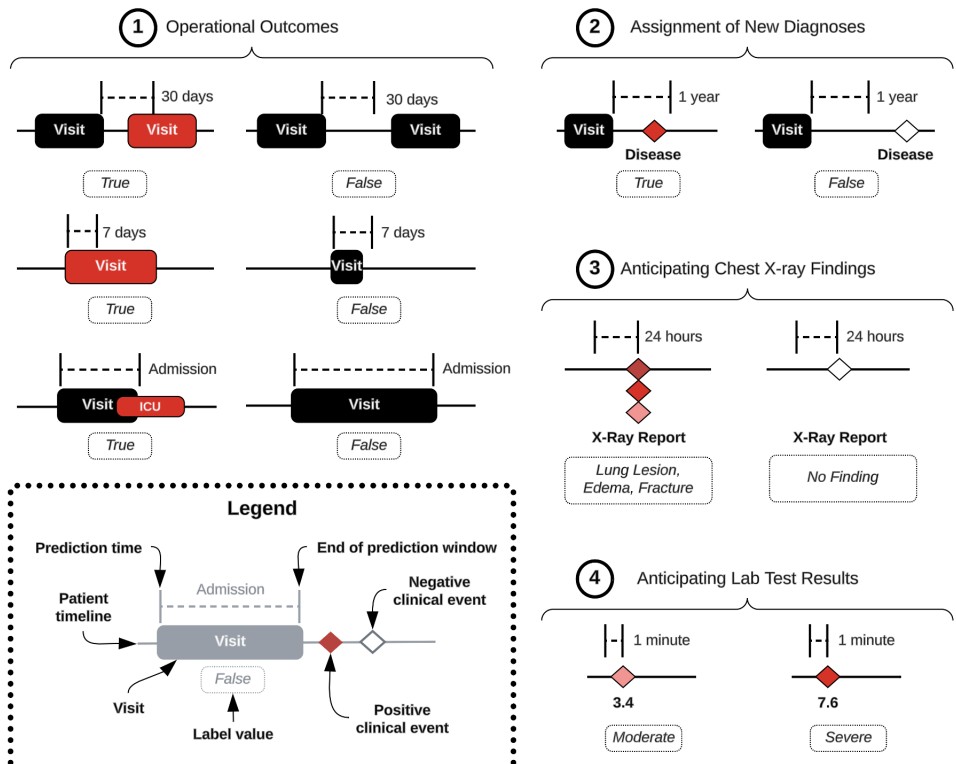

Figure 2: Summary of Benchmark Tasks. Each subfigure contains one of the 4 types of predictive classification tasks included in our benchmark: (1) *Operational Outcomes* (binary), (2) *Assignment of New Diagnoses* (binary), (3) *Anticipating Chest X-ray Findings* (multilabel), (4) *Anticipating Lab Test Results* (multiclass). Each **black line** represents a patient timeline. The black boxes represent how each timeline would be labeled for each task at a specific prediction time. The leftmost edge of the dotted lines above each timeline is the prediction time, and the rightmost edge is the end of the time horizon for that task. Note that each *Operational Outcome* task (1) has a different prediction window (30 days, 7 days, duration of admission), while the other three task categories (2, 3, 4) all have uniform prediction windows across their subtasks.

We chose these two models as our baselines for several reasons. First, language modeling has achieved state-of-the-art results on clinical prediction tasks [41, 32, 24, 28, 19], while count-based featurization remains a simple but competitive baseline [31, 32, 41]. Second, most prior FMs trained on structured EHR data have not had their model weights published, and were developed and tested exclusively on nonstandard data formats like MIMIC-III [48]. This makes it nearly impossible to conduct a fair comparison of prior models, which often requires re-implementation or significant modification to work across datasets [13]. This is one of the key challenges we are attempting to solve with EHRSHOT. We pre-train our own FM from scratch to have full control over its training, and publish its model weights so the community can reproduce and build upon our results.

**Count-based Features**. Count-based featurization is a well-established baseline for EHR tasks, valued for its simplicity and effectiveness [31]. The fundamental idea involves converting each patient's timeline into a count vector, where each element contains the number of occurrences of a specific medical concept prior to the prediction time of a task. These patient vectors are combined into a count matrix, which is high-dimensional and sparse. We use a technique called *ontology expansion* to increase the density of representation and improve the accuracy of code coverage by acknowledging the parent/child hierarchical relationships between medical concepts [4]. After generating our ontology-expanded count matrix, we train a gradient boosting machine (GBM) model on the EHRSHOT train split, and tune hyperparameters on the validation split. We use the LightGBM implementation [17]. We also evaluate a Logistic Regression and Random Forest model as baselines.

Table 3: Task Demographics. The number of unique patients and total labels for each task. A single patient may have multiple labels for one task (e.g. a patient with multiple anemia lab results). We show the prevalence of positive patients/labels in parenthesis. For the multiclass lab test tasks, we define a positive label as any non-normal result. For the multilabel chest X-ray task, we define a positive label as a report with at least one finding.

| Task Name | Train | | Val | | Test | |
|---|---|---|---|---|---|---|
| | # Patients (# Positive) | # Labels (# Positive) | # Patients (# Positive) | # Labels (# Positive) | # Patients (# Positive) | # Labels (# Positive) |
| **Operational Outcomes** | | | | | | |
| Long Length of Stay | 1377 (464) | 2569 (681) | 1240 (395) | 2231 (534) | 1238 (412) | 2195 (552) |
| 30-day Readmission | 1337 (164) | 2609 (370) | 1192 (159) | 2207 (281) | 1190 (151) | 2189 (260) |
| ICU Transfer | 1306 (107) | 2402 (113) | 1157 (84) | 2052 (92) | 1154 (75) | 2037 (85) |
| **Anticipating Lab Test Results** | | | | | | |
| Thrombocytopenia | 2084 (870) | 68776 (9774) | 1981 (774) | 54504 (6962) | 1998 (818) | 56338 (7960) |
| Hyperkalemia | 2038 (383) | 76349 (1215) | 1935 (348) | 60168 (886) | 1958 (339) | 63653 (948) |
| Hypoglycemia | 2054 (422) | 122108 (1065) | 1950 (362) | 95488 (858) | 1970 (356) | 100568 (783) |
| Hyponatremia | 2035 (1288) | 81336 (20181) | 1930 (1165) | 64473 (14674) | 1956 (1212) | 67028 (16003) |
| Anemia | 2092 (1251) | 70501 (9544) | 1992 (1122) | 56224 (7445) | 2002 (1151) | 58155 (7636) |
| **Assignment of New Diagnoses** | | | | | | |
| Hypertension | 793 (130) | 1260 (184) | 784 (130) | 1250 (177) | 758 (130) | 1261 (160) |
| Hyperlipidemia | 923 (137) | 1684 (205) | 863 (140) | 1441 (189) | 864 (133) | 1317 (172) |
| Pancreatic Cancer | 1376 (128) | 2576 (155) | 1242 (46) | 2215 (53) | 1246 (40) | 2220 (56) |
| Celiac | 1392 (48) | 2623 (62) | 1252 (8) | 2284 (11) | 1255 (13) | 2222 (21) |
| Lupus | 1377 (79) | 2570 (104) | 1239 (24) | 2226 (33) | 1249 (19) | 2243 (20) |
| Acute MI | 1365 (130) | 2534 (175) | 1234 (112) | 2177 (146) | 1235 (115) | 2127 (144) |
| **Anticipating Chest X-ray Findings** | | | | | | |
| Chest X-Ray Findings | 251 (237) | 7481 (4771) | 395 (378) | 9366 (6032) | 399 (381) | 9428 (6400) |

Their results can be seen in Appendix in Figures 10 and 11. For clarity, we exclude them from the following analyses, as they perform roughly at par with the count-based GBM model.

**Clinical Language-Model-Based Representations using Transformers (CLMBR-T-base).** CLMBR-T-base is an autoregressive model designed to predict the next medical code in a patient's timeline given previous codes. This objective enables it to learn robust global patterns for clinical prediction tasks. It is based on the CLMBR model originally developed in [41], but following [8] we substitute a transformer in place of a GRU as its base model. Our model employs causally masked local attention. This ensures forward-only flow of information which is vital for prediction tasks, and is in contrast to BERT-based models which are bidirectional in nature [41]. Note that our model does not process clinical text, only structured information. Our model has 141M trainable parameters, a hidden dimension of 768, and a next code prediction objective. This provides our version of CLMBR-T-base with minute-level resolution rather than the day-level aggregation of the original model formulation [41]. We leave training larger versions of CLMBR to future work.

More details about our baseline models can be found in the Appendix in Section D.

## 5 Results

We evaluate each baseline model in a few-shot setting. For each of the 15 benchmark tasks, we steadily increase the number of examples $k$ that each model sees from $k = 1$ to the full training dataset, and record the model's AUROC and AUPRC at each $k$.

More precisely, we define "$k$-shot evaluation" of a model $M$ on a specific task $T$ as follows. We train $M$ on $k$ positive examples and $k$ negative examples sampled from $T$'s training split. We then select an additional $k$ positive examples and $k$ negative examples from $T$'s validation split, and use these validation examples to select the best hyperparameters for $M$ for task $T$. Finally, we evaluate the AUROC and AUPRC of the best performing version of $M$ on $T$'s entire held-out test split. For tasks where the total number of unique positive examples is less than $k$, we include all positive examples in our training set, and randomly resample positive examples until the total number of training examples seen by the model is $k$. We consider values of $k \in \{1, 2, 4, 8, 12, 16, 24, 32, 48, 64, 128\}$ for all tasks (with the exception of Celiac, for which we limit $k \leq 64$ as there are only 62 positive training labels).

For the count-based GBM, these few-shot examples are the only training examples seen by the model. For the pretrained CLMBR-T-base model, we use these few-shot examples to fine-tune a logistic regression head appended to the top of the model, while keeping the weights of the pretrained CLMBR-T-base model frozen. Pretraining the CLMBR-T-base model took roughly 4 days on a single Nvidia V100 hosted in an on-premise compute cluster.

The AUROC of each model across all 4 task categories is presented in Figure 3. In the Appendix, we show this grouping for AUPRC in Figure 5. We also break down each individual task's AUROC in Figure 6 and AUPRC in Figure 7 of the Appendix. We also include results for additional baselines in the Appendix in Figures 10 and 11. The **bolded lines** are the Macro-AUC for each model within a task category, averaged across all subtasks at each $k$. We include the performance of each model trained on the entire EHRSHOT training split on the far right of every plot as *All*.

As shown in Figure 3, the pretrained foundation model CLMBR-T-base (blue) outperforms the count-based GBM (red) across all aggregated task categories for $k \leq 64$. This demonstrates the benefits of pretraining in few-shot settings, as the model can leverage patterns learned across millions of patients to derive more accurate representations out-of-the-box than a model trained from scratch. CLMBR-T-base outperforms the count-based GBM across all $k$ on the *Operational Outcomes* and the majority of *Anticipating Lab Test Results* and *Anticipating Chest X-ray Findings* tasks. For these three task groups, the advantage of CLMBR-T-base seems most pronounced at intermediate levels of $k$ between 8 and 128. At extremely low $k$ (i.e. $k = 1$), both models struggle to learn anything, while as $k$ increases the advantage of the pretrained model tends to shrink, a trend noted elsewhere [21]. This is most visible in the far-right of the plot at the *All* marker, which represents the performance of each model when trained on the full EHRSHOT training dataset.

In fact, the count-based GBM exceeds the performance of CLMBR-T-base on the *Assignment of New Diagnoses* tasks at $k > 64$. This suggests that the advantage of pretraining comes primarily from improved initialization of patient representations, and that the largest gains are achieved in the most data poor regimes.

There are several possible reasons for CLMBR-T-base's underperformance at higher values of $k$ for the *Assignment of New Diagnoses* tasks. First, the CLMBR-T-base model's training objective is next code prediction, which makes it ill-suited for predictive tasks with long time horizons (which for these tasks is 1 year). Second, if a simple tree-based model exists for a task (i.e. a few medical concepts tightly correlate with a diagnosis), then it may be more difficult for a pretrained model to coerce patient representations learned over millions of patients to that specific task than training a model from scratch with enough data to learn those distinctive signals. We believe that this reversal in model rankings demonstrates a key strength of EHRSHOT – namely, the diversity of its predictive tasks can help identify opportunities for improving pretraining and few-shot strategies.

We release all of our model weights, evaluation tasks, and data processing code to fully reproduce our results. To the best of our knowledge, the release of our pretrained CLMBR-T-base model is one of the first examples of such a clinical FM having its pretrained weights made publicly available [48].

# 6 Discussion

We believe that EHRSHOT represents a useful contribution to the ML community by enabling more reproducible healthcare ML research. The release of our pretrained CLMBR-T-base model's weights will allow the community to replicate and build upon our work. Our results identify opportunities for improving pretrained models in few-shot settings.

Acquiring labeled EHR data is expensive and time-consuming. Additionally, certain rare conditions may only be present in a small cohort of patients out of millions within a health system [28]. Thus, model performance in low-label settings is of paramount importance in healthcare. As our results in Section 5 demonstrate, pretrained FMs can yield large performance gains in few-shot settings. While we acknowledge that the tasks themselves may not be the most clinically meaningful, we believe that EHRSHOT offers a valuable contribution by providing a reproducible and rigorous point of comparison for different technical approaches to developing clinical FMs.

**Limitations**. There are several limitations to this work. First, we only release structured data – i.e. we do not publish any of the clinical text or images associated with our patients. While many datasets for medical images exist [3], publishing clinical text remains a challenge [40]. Second, we only

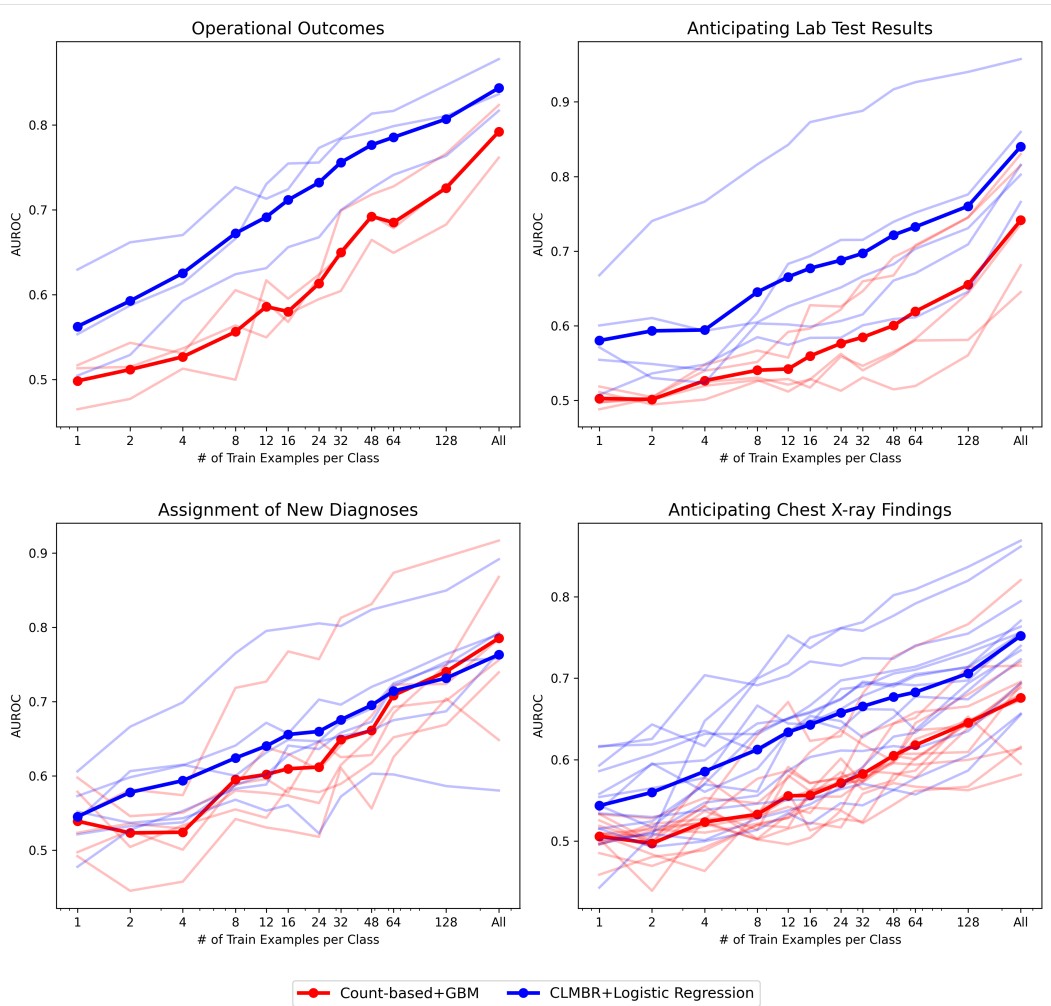

Figure 3: Aggregated AUROC across all subtasks within each of the 4 task categories for $k \in \{1, 2, 4, 8, 12, 16, 24, 32, 48, 64, 128\}$ shots. We show performance on the full training set as *All*. The **bolded lines** are the Macro-AUROC for each model, averaged across all subtasks within a task category for each value of $k$. The blurred lines are the average AUROC across 5 replicates for each subtask within a task category. CLMBR-T-base (blue) consistently outperforms the count-based GBM (red) at $k \leq 64$, but lags in higher label settings for the *Assignment of New Diagnoses* tasks.

consider one type of foundation model (CLMBR-T-base) for our experiments [41]. We look forward to seeing the additional foundation models that the community applies to our benchmark. Third, we release a very small cohort of patients (<1%) from our source EHR database, and specifically select these patients for the tasks that we define. Releasing our full pretraining dataset would be infeasible from a governance and effort perspective. Thus, while necessary in order to publish our EHR dataset and still broader than existing ICU-specific datasets, our cohort selection process limits the types of questions we can answer and does not reflect the full diversity of medical data. Fourth, as we only were able to evaluate our pretrained FM on Stanford Medicine data, it is unclear how well our pretrained model will perform at other institutions. We anticipate there will be some drop in performance, but the extent is unclear. Fifth, several of our tasks are "low label" in the most extreme sense – for example, the Celiac task only has 13 positive patients in its test set. This makes obtaining low variance estimates of model performance difficult. We aim to mitigate this by adding additional patients to our benchmark in future releases.

**Societal Implications**. We believe that the release of this dataset can help spur positive innovations for improving clinical care with ML. However, we recognize that there are patient privacy concerns anytime EHR data is released. We believe we sufficiently mitigate this risk through the rigorous deidentification process on which our data is subjected [5]. Additionally, we gate access to the dataset through a research data use agreement. Another concern is that models trained on biased data will reflect those biases [7]. Thus, the pretrained FM that we release may propagate biases in care delivery or outcomes present in our source EHR database [7]. However, we hope that by encouraging the full release of models, we can help the community better identify and mitigate these issues [25].

## 7   Conclusion

We publish EHRSHOT, a benchmark containing the structured data of 6,739 patients' full longitudinal medical timelines specifically geared towards few-shot evaluation of foundation models for clinical data. Unlike most prior work, EHRSHOT contains longitudinal health data rather than a single department (e.g. ICU). We define a set of 15 tasks ranging from well-studied outcomes like 30-day readmission to lesser explored settings such as anticipating abnormal lab values. Finally, we release the weights of a foundation model pretrained on over 2.57M patient timelines and publish the code needed to replicate our results. We hope that this work represents a first step towards moving the field of ML for healthcare towards more reproducible and open model development.

## Acknowledgments and Disclosure of Funding

We thank the Stanford AIMI Center and Stanford Medicine Research IT for their assistance in publishing this dataset. This work was supported in part by the Mark and Debra Leslie Endowment for AI in Healthcare, the Clinical Excellence Research Center at Stanford Medicine, and Technology and Digital Solutions at Stanford Healthcare. MW is supported by an NSF Graduate Research Fellowship. JF was supported in part by a Stanford AIMI-HAI Partnership Grant.

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
