# Supplementary Material

## A  Author Responsibility Statement

The authors confirm that they bear all responsibility in case of violation of rights or licenses.

## B  Public Accessibility + Licenses

### B.1  Dataset

We release EHRSHOT under a research data use agreement. The dataset is available here: https://ehrshot.stanford.edu/. Access is gated by a researcher data use agreement due to the sensitive nature of the dataset. We do not upload our dataset to another data repository due to these concerns.

In order to ensure we do not reveal Protected Health Information (PHI) in our dataset, we take several precautions. First, we only release deidentified data. The deidentification process has been previously described in [5]. Second, on top of this deidentification process, we also apply additional privacy-protecting transformations following the best practices of the MIMIC-III dataset [16], which are detailed in Section C.5. Third, we do not publish any clinical notes. Fourth, we release our dataset under a data usage agreement that requires researchers to register with their identity and gain approval before accessing the dataset.

**License:** The license for the dataset is the standard Stanford University Dataset Research Use Agreement, and is reproduced below:

By registering for downloads from the EHRSHOT Dataset, you are agreeing to this Research Use Agreement, as well as to the Terms of Use of the Stanford University School of Medicine website as posted and updated periodically at http://www.stanford.edu/site/terms/. Permission is granted to view and use the EHRSHOT Dataset without charge for personal, non-commercial research purposes only. Any commercial use, sale, or other monetization is prohibited.

Other than the rights granted herein, the Stanford University School of Medicine ("School of Medicine") retains all rights, title, and interest in the EHRSHOT Dataset. You may make a verbatim copy of the EHRSHOT Dataset for personal, non-commercial research use as permitted in this Research Use Agreement. If another user within your organization wishes to use the EHRSHOT Dataset, they must register as an individual user and comply with all the terms of this Research Use Agreement. YOU MAY NOT DISTRIBUTE, PUBLISH, OR REPRODUCE A COPY of any portion or all of the EHRSHOT Dataset to others without specific prior written permission from the School of Medicine. YOU MAY NOT SHARE THE DOWNLOAD LINK to the EHRSHOT Dataset to others. If another user within your organization wishes to use the EHRSHOT Dataset, they must register as an individual user and comply with all the terms of this Research Use Agreement. You must not modify, reverse engineer, decompile, or create derivative works from the EHRSHOT Dataset. You must not remove or alter any copyright or other proprietary notices in the EHRSHOT Dataset.

### B.2  Pretrained Foundation Model (CLMBR-T-base)

We release CLMBR-T-base, a foundation model pre-trained on the structured EHR data of roughly 2.5 million patients at Stanford Medicine [41]. The model's weights can be found at our website here: https://ehrshot.stanford.edu/. Access is gated by a researcher data use agreement due to the sensitive nature of the training dataset.

A concern with the release of such a model is the lack of solid theoretical privacy assurances, thus creating the possibility of the model revealing medical data. To mitigate these concerns, we implement several additional precautions. First, the model is trained exclusively on deidentified data to eliminate the chance of any Protected Health Information (PHI) seeping into the model. Second, all unique text strings released as part of our CLMBR-T-base model's dictionary (e.g. terms such as "Yes" or "No" that serve as categorical variables) were manually reviewed to ensure they do not reveal any

PHI. Third, we make our model available under a data usage agreement that requires researchers to register with their identity and gain approval before accessing the model.

**License:** The license for the code for the model is here: https://github.com/som-shahlab/femr/blob/main/LICENSE. The license for the model weights is here: https://huggingface.co/StanfordShahLab/clmbr-t-base.

## C Dataset Details

### C.1 EHRSHOT Cohort

Demographics of the EHRSHOT cohort are included below.

Table 4: EHRSHOT: Patient demographics in the train, validation, and test splits.

| Attribute | | Train | Val | Test | All Splits |
|---|---|---|---|---|---|
| **Gender** | Male | 1122 | 1090 | 1086 | 3298 |
| | Female | 1173 | 1142 | 1126 | 3441 |
| | 19-20 | 8 | 3 | 2 | 13 |
| | 21-40 | 412 | 457 | 431 | 1300 |
| **Age** | 41-60 | 648 | 597 | 576 | 1821 |
| | 61-80 | 916 | 892 | 905 | 2713 |
| | 81-88 | 311 | 283 | 298 | 892 |
| | American Indian | 14 | 7 | 4 | 25 |
| | Asian | 356 | 347 | 340 | 1043 |
| | Black | 98 | 105 | 95 | 298 |
| **Race** | Pacific Islander | 23 | 21 | 30 | 74 |
| | White | 1286 | 1222 | 1228 | 3736 |
| | Unknown | 518 | 530 | 515 | 1563 |
| **Ethnicity** | Hispanic | 374 | 342 | 322 | 1038 |
| | Non-Hispanic | 1921 | 1890 | 1890 | 5701 |
| **Total** | | 2295 | 2232 | 2212 | 6739 |

### C.2 Pretraining Dataset

The pretraining dataset for CLMBR-T-base contains a total of 3.67 million patient records, of which 2.57 million are used to train the model. We include summary statistics of these patients' demographics in Table 5 and Table 6.

### C.3 Task Definitions

Here, we detail the precise definitions for each of the 15 tasks for which we provide labels in our benchmark dataset.

**Operational Outcomes**. These tasks are related to hospital operations. They are all binary classification tasks, and are defined as follows:

- **Long Length of Stay**: Predict whether a patient's total length of stay during a visit to the hospital will be at least 7 days. The prediction time is at 11:59pm on the day of admission, and visits that last less than one day (i.e. discharge occurs on the same day of admission) are ignored.

- **30-day Readmission**: Predict whether a patient will be re-admitted to the hospital within 30 days after being discharged from a visit. The prediction time is at 11:59pm on the day of admission, and admissions where a readmission occurs on the same day as the corresponding discharge are ignored.

Figure 4: Histograms of EHRSHOT patient timeline characteristics

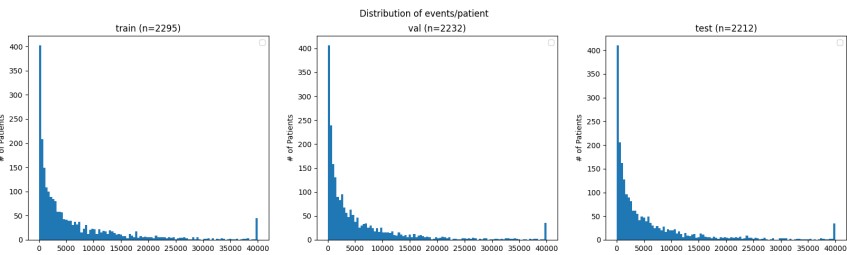

(a) Total number of events per patient, broken down by train/val/test split. Note that the x-axis is clamped at 40000 for clarity (i.e. along the x-axis we plot $\min(x, 40000)$)

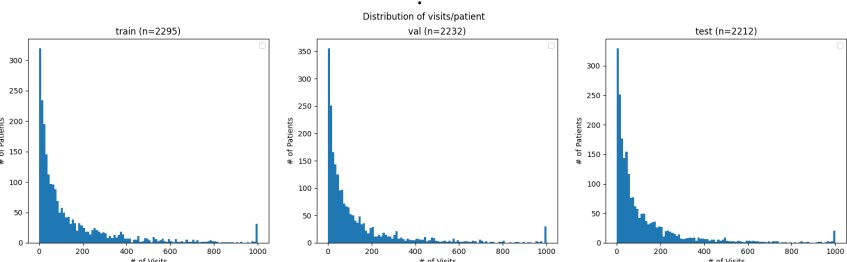

(b) Total number of visits per patient, broken down by train/val/test split. Note that the x-axis is clamped at 1000 for clarity (i.e. along the x-axis we plot $\min(x, 1000)$)

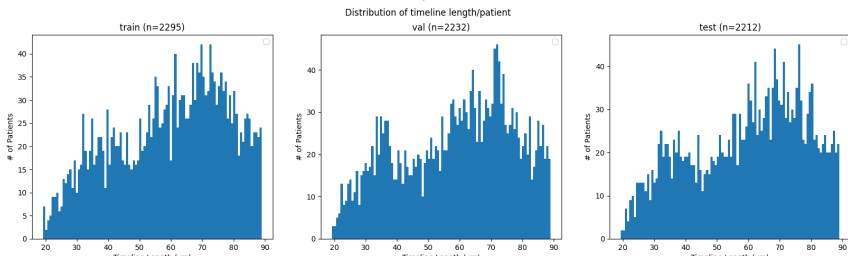

(c) Total length of each patient timeline (i.e. difference in time between birth date and last recorded event), broken down by train/val/test split.

- **ICU Transfer**: Predict whether a patient will be transferred to the ICU during a visit to the hospital. The prediction time is at 11:59pm on the day of admission, and ICU transfers that occur on the same day as admission are ignored.

**Anticipating Lab Test Results**. These tasks are related to lab value prediction. They are all multiclass classification tasks. The prediction time is immediately before the lab result is recorded. They are defined as follows:

- **Thrombocytopenia**: Predict whether a thrombocytopenia lab comes back as normal (>=150 $10^9$/L), mild (>=100 and <150 $10^9$/L), moderate (>=50 and <100 $10^9$/L), or severe (<50 $10^9$/L),. We consider all lab results coded as LOINC/LP393218-5, LOINC/LG32892-8, or LOINC/777-3.
- **Hyperkalemia**: Predict whether a hyperkalemia lab comes back as normal (<=5.5 mmol/L), mild (>5.5 and <=6mmol/L), moderate (>6 and <=7 mmol/L), or severe (>7 mmol/L). We consider all lab results coded as LOINC/LG7931-1, LOINC/LP386618-5, LOINC/LG10990-6, LOINC/6298-4, or LOINC/2823-3.
- **Hypoglycemia**: Predict whether a hypoglycemia lab comes back as normal (>=3.9 mmol/L), mild (>=3.5 and <3.9 mmol/L), moderate (>=3 and <3.5 mmol/L), or severe (<3 mmol/L). We consider all lab results coded as SNOMED/33747003, LOINC/LP416145-3, or LOINC/14749-6.

Table 5: Pretraining Dataset: Patient demographics in the train, validation, and test splits.

| Attribute | | Train | Val | Test | All Splits |
|---|---|---|---|---|---|
| **Gender** | Male | 1186614 | 255179 | 254733 | 1696526 |
| | Female | 1380836 | 295126 | 296127 | 1972089 |
| **Age** | 0-20 | 625949 | 135045 | 134684 | 895678 |
| | 21-40 | 671018 | 143607 | 144045 | 958670 |
| | 41-60 | 617519 | 131996 | 132432 | 881947 |
| | 61-80 | 502842 | 107299 | 107746 | 717887 |
| | 81-88 | 150122 | 32358 | 31953 | 214433 |
| **Race** | American Indian | 7229 | 1509 | 1516 | 10254 |
| | Asian | 371065 | 79638 | 80418 | 531121 |
| | Black | 83624 | 17895 | 17919 | 119438 |
| | Pacific Islander | 20959 | 4350 | 4435 | 29744 |
| | White | 987676 | 211262 | 211429 | 1410367 |
| | Unknown | 1096897 | 235651 | 235143 | 1567691 |
| **Ethnicity** | Hispanic | 325037 | 69912 | 69689 | 464638 |
| | Non-Hispanic | 2242413 | 480393 | 481171 | 3203977 |
| **Total** | | 2567450 | 550305 | 550860 | 3668615 |

Table 6: Pretraining Dataset: Summary statistics on the number of events, visits, and length of patient timelines.

| Attribute | | Train | Val | Test | All Splits |
|---|---|---|---|---|---|
| **Number of Events** | Min | 1 | 1 | 1 | 1 |
| | Mean | 707 | 706 | 704 | 706 |
| | Max | 191369 | 213133 | 214400 | 214400 |
| **Number of Visits** | Min | 0 | 0 | 0 | 0 |
| | Mean | 28 | 28 | 28 | 28 |
| | Max | 3701 | 4305 | 3109 | 4305 |
| **Timeline Length (yrs)** | Min | 0 | 0 | 0 | 0 |
| | Mean | 40 | 40 | 40 | 40 |
| | Max | 92 | 90 | 90 | 92 |

- **Hyponatremia**: Predict whether a hyponatremia lab comes back as normal (>=135 mmol/L), mild (>=130 and <135 mmol/L), moderate (>=125 and <130 mmol/L), or severe (<125 mmol/L). We consider all lab results coded as LOINC/LG11363-5, LOINC/2951-2, or LOINC/2947-0.

- **Anemia**: Predict whether an anemia lab comes back as normal (>=120 g/L), mild (>=110 and <120 g/L), moderate (>=70 and <110 g/L), or severe (<70 g/L). We consider all lab results coded as LOINC/LP392452-1.

Please note that for the results of our baseline experiments in Section 5, we reframe these lab value tasks as binary classification tasks, where a label is "negative" if the result is normal and "positive" otherwise.

**Assignment of New Diagnoses**. These tasks are related to predicting the first diagnosis of a disease. They are all binary classification tasks. The prediction time is at 11:59pm on the day of discharge from an inpatient visit, and we count any diagnosis that occurs within 365 days post-discharge as a positive outcome. We ignore all discharges in which the patient already has an existing diagnosis of a disease. The tasks are defined as follows:

Table 7: Task Prediction Windows. *Prediction Time* is the precise time point (up to minute precision) in a patient's timeline when the prediction is made. *Time Horizon* is the length of time considered after the prediction time to determine whether an event occurs, i.e. we only consider a patient "positive" for a new diagnosis of pancreatic cancer if she receives that diagnosis within a year of being discharged.

| Task Name | Task Type | Prediction Time | Time Horizon |
|---|---|---|---|
| **Operational Outcomes** | | | |
| Long Length of Stay | Binary | 11:59pm on day of admission | Admission duration |
| 30-day Readmission | Binary | 11:59pm on day of discharge | 30 days post-discharge |
| ICU Transfer | Binary | 11:59pm on day of admission | Admission duration |
| | | | |
| **Anticipating Lab Test Results** | | | |
| Thrombocytopenia | 4-way multiclass | Immediately before result | Next result |
| Hyperkalemia | 4-way multiclass | Immediately before result | Next result |
| Hypoglycemia | 4-way multiclass | Immediately before result | Next result |
| Hyponatremia | 4-way multiclass | Immediately before result | Next result |
| Anemia | 4-way multiclass | Immediately before result | Next result |
| | | | |
| **Assignment of New Diagnoses** | | | |
| Hypertension | Binary | 11:59pm on day of discharge | 1 year post-discharge |
| Hyperlipidemia | Binary | 11:59pm on day of discharge | 1 year post-discharge |
| Pancreatic Cancer | Binary | 11:59pm on day of discharge | 1 year post-discharge |
| Celiac | Binary | 11:59pm on day of discharge | 1 year post-discharge |
| Lupus | Binary | 11:59pm on day of discharge | 1 year post-discharge |
| Acute MI | Binary | 11:59pm on day of discharge | 1 year post-discharge |
| | | | |
| **Anticipating Chest X-ray Findings** | | | |
| Chest X-Ray Findings | 14-way multilabel | 24hrs before report is recorded | Next report |

- **Hypertension**: Predict whether the patient will have her first diagnosis of essential hypertension within the next year. We define hypertension as an occurrence of the code SNOMED/59621000, as well as its children codes in our ontology.

- **Hyperlipidemia**: Predict whether the patient will have her first diagnosis of hyperlipidemia within the next year. We define hyperlipidemia as an occurrence of the code SNOMED/55822004, as well as its children codes in our ontology.

- **Pancreatic Cancer**: Predict whether the patient will have her first diagnosis of pancreatic cancer within the next year. We define pancreatic cancer as an occurrence of the code SNOMED/372003004, as well as its children codes in our ontology.

- **Celiac**: Predict whether the patient will have her first diagnosis of celiac disease within the next year. We define celiac disease as an occurrence of the code SNOMED/396331005, as well as its children codes in our ontology.

- **Lupus**: Predict whether the patient will have her first diagnosis of lupus within the next year. We define lupus as an occurrence of the code SNOMED/55464009, as well as its children codes in our ontology.

- **Acute MI**: Predict whether the patient will have her first diagnosis of an acute myocardial infarction within the next year. We define myocardial infarction as an occurrence of the code SNOMED/57054005, as well as its children codes in our ontology.

**Anticipating Chest X-ray Findings**. The chest X-ray findings task is a multilabel classification task to identify which of 14 possible findings were included in a chest X-ray report. The prediction time is 24 hours before the radiology report is recorded. The labels are derived by running the CheXpert

NLP labeler on the unstructured text of the corresponding radiology report [14]. We do not release this unstructured text as part of our dataset due to patient privacy concerns.

The possible findings are as follows: "No Finding", "Enlarged Cardiomediastinum", "Cardiomegaly", "Lung Lesion", "Lung Opacity", "Edema", "Consolidation", "Pneumonia", "Atelectasis", "Pneumothorax", "Pleural Effusion", "Pleural Other", "Fracture", "Support Devices".

## C.4 Dataset Format

Our dataset is comprised of two main sets of tabular files: **(A) Events** files which contain all of the clinical events associated with every patient in our dataset, and **(B) Labels** files which contain the labels associated with all of our benchmark tasks for every patient in our dataset.

**(A) Events** is as a set of CSV files containing every clinical event that happened to the patients in our dataset. Every row is a unique clinical event. Each CSV file shares the same column schema, which is as follows:

- **Patient ID** - Integer - Unique identifier for patient
- **Start** - Datetime - Start time of event
- **End** - Datetime (optional) - End time of event
- **Code** - String - Name of the clinical event (e.g. "SNOMED/3950001" or "ICD10/I25.110")
- **Value** - Float/String (optional) - Either a numerical value associated with an event (e.g. a lab test result) or a string associated with a categorical variable (e.g. "Yes/No" questions)
- **Unit** - String (optional) - Unit of measurement for **Value**
- **Visit ID** - Integer (optional) - Unique identifier for the visit during which this event occurred
- **OMOP-CDM Table** - String - Name of the source OMOP-CDM table where this event was recorded

Every event is associated with the OMOP-CDM table in the source STARR database from which it was taken (**OMOP-CDM Table**) [34]. Researchers unfamiliar with the OMOP-CDM can simply ignore this column.

**(B) Labels** is a set of CSV files containing the labels for every task for every patient. Every row is a unique label associated with a specific patient, task, and time point. Each CSV file shares the same column schema, which is as follows:

- **Patient ID** - Integer - Unique identifier for patient
- **Prediction Time** - Datetime - Time at which the prediction for this label is made
- **Value** - Boolean / Integer - Value for this label. Boolean if task is binary classification. Integer if task is multiclass or multilabel classification.
- **Label Type** - String - Type of task associated with this label. Can be "boolean" (binary classification), "numeric" (regression), "survival" (time-to-event), or "categorical" (multilabel or multiclass classification).

## C.5 Data Preprocessing

The source dataset we use, the Stanford STARR research database [5], is an Observational Medical Outcomes Partnership Common Data Model (OMOP-CDM) [12] compliant transformation of data extracted from Stanford's production EHR system (Epic). We do not alter any of the transformations or deidentification steps in the ETL used to generate this OMOP-CDM extract described in [5].

Following the best practices of the MIMIC-III dataset, we apply several additional custom transformations to prevent data leakage and add an additional layer of patient privacy protections [16]. First, we jitter all dates within each patient timeline by the same random amount (to a random year between 2100 and 2200). Second, we remove all patients <= 18 or >= 89 years of age. Third, we remove all instances of free form text (i.e., notes and narratives). For the clinical events which take on categorical values specified as strings (e.g. a questionnaire which can be answered "Yes" or "No"), we select the top-100 most representative such categorical text strings, manually verify that they do

not contain any PHI, and remove the rest of the text strings from our model release by replacing them with blank strings. This preserves roughly 65% of all categorical values in our dataset. Fourth, we remove any patients with less than 10 clinical events in their record. Fifth, we adjust the timing of certain events to more realistically reflect the chronology of care delivery. Specifically, we move any events recorded before a patient's birth to after their time of birth; we set the start times of visits equal to the start time of the first event in each visit; we move billing codes recorded during a visit to the end of the visit; we move any event coded at midnight to 11:59pm of that day; we remove all duplicate codes that occur sequentially on the same day; and we remove all codes with 'None' values that occur on the same day as an identical code with a non-'None' value associated with it. These transformations are all specified in code in our Github repo.

## C.6 Cohort Selection Process

We selected a cohort of 6,739 patients for EHRSHOT from the larger STARR source dataset of 3.67M patients. Per the motivation of this project, we were primarily interested in few-shot evaluation of models across diverse tasks. Several of the tasks that would be of interest to a health system, however, have fairly low prevalence within the general patient population. Thus, we needed to construct our cohort in a way that preserved sufficient positive labels to enable downstream models to conduct few-shot learning. We aimed to have at least $k = 128$ positive and negative examples in each of the train/val/test splits for every task that we considered in order to allow for a broad range of few-shot learning scenarios, and at least $k = 128$ positive examples for each label within a multiclass or multilabel classification task. Where this was not possible (e.g. the Celiac task), we included as many positive labels in each split as possible.

We began with our set of 15 tasks of interest. For each task, we labeled all patients within our source database per that task's definition. For tasks that have a low prevalence (which we consider as a 1:5 ratio of positive to negative labels), we subsample negative labels to bring the prevalence of positive labels up to that ratio. We then subsample further for few-shot evaluation, selecting 128 unique patients for each split who have at least one positive label for the task. We then sample sufficient negative labels to maintain the chosen prevalence. We repeat this process for all tasks to arrive at our final cohort of patients. For each successive task, we prioritize selecting patients that have already been sampled into our cohort to reduce the total number of patients added to our cohort (since some patients have positive labels for multiple tasks).

# D Results Details

In order to fully reproduce our results, please follow the instructions at our Github repo here: https://github.com/som-shahlab/ehrshot-benchmark.

## D.1 Problem Formulation

Our dataset and models can be formulated as follows. Our dataset $\mathcal{D} = (\{(\mathbf{X}_p, \mathbf{Y}_p)\}_{p=1}^{|\mathcal{P}|}$ contains the full coded medical timeline ($\mathbf{X}_p$) and task-specific set of labels ($\mathbf{Y}_p$) for each patient $p \in \mathcal{P}$, for a total of $|\mathcal{P}|$ patients. Each patient $p$ is defined by a sequence of clinical events $\mathbf{X}_p = \{x_{p1}, x_{p2}, ..., x_{pn}\}$, where $x_{pi}$ denotes the $i$th code in the timeline of patient $p$. Note that a code $x_{pi}$ can be any form of structured data taken from the patient's EHR, including a diagnosis, procedure, medication prescription, lab test, etc. We define $\mathbf{X}_p^{(t)}$ to be the patient timeline up to time $t$ – i.e. if event $x_{pj}$ occurs before or at $t$ but $x_{p(j+1)}$ occurs after $t$, then if $\mathbf{X}_p = \{x_{p1}, ..., x_{pj}, x_{p(j+1)}, ..., x_{pn}\}$ we have that $\mathbf{X}_p^{(t)} = \{x_{p1}, ..., x_{pj}\}$.

In addition to the timeline of each patient, our dataset also contains labels for each task and patient. We define benchmark tasks $b \in \mathcal{B}$, where $|\mathcal{B}| = 15$ for our dataset. Each patient has a set of labels $\mathbf{Y}_p = \{y_{pb_1}^{(t_1)}, y_{pb_1}^{(t_2)}, ..., y_{pb_{|\mathcal{B}|}}^{(t_L)}\}$, where $L$ is the total number of labels for patient $p$, and the expression $y_{pb_i}^{(t_j)}$ represents the label for patient $p$ for task $b_i$ at time point $t_j$.

We are interested in making predictions of the following format: Given a patient $p$'s entire medical history up to and including time point $t$ (i.e. $\mathbf{X}_p^{(t)}$), predict the value of $y_{pb}^{(t)}$ for each corresponding benchmark task $b \in \mathcal{B}$ where such a label exists.

Please note that this prediction task is at the level of individual clinical events rather than visits/encounters.

## D.2    Count-Based GBM

We train a LightGBM model as one of our baseline models. In order to train such a model on a patient's timeline, we must first featurize the timeline into a vector. We follow best practices for competitive baseline models by using count-based featurization, in which a patient is transformed into a vector containing the counts of how many times each clinical event has occurred in that patient's timeline prior to the prediction time point [31, 41].

Let $\mathcal{C}$ be the set of all unique medical codes in our dataset. Let us consider making a prediction for patient $p$ at time $t$. Then the count-based featurization for $p$ at time $t$ is given by the vector $\mathbf{p}^{(t)} \in \mathbb{N}^{|\mathcal{C}|}$, where each element is defined as $\mathbf{p}_i^{(t)} = \sum_{x_j \in \mathbf{X}_p^{(t)}} I(x_j = i)$, i.e. the count of medical code $i$ recorded for patient $p$ before the prediction time $t$. Stacking these patient vectors results in a count matrix $\mathbf{M} \in \mathbb{N}^{|\mathbf{Y}| \times |\mathcal{C}|}$. As there are hundreds of thousands of unique codes, most of which occur infrequently among patients, this results in a very high-dimensional and sparse matrix.

To help address the sparseness of $\mathbf{M}$, we use a technique called *ontology expansion* [4], in which we count each occurrence of a code once for the code itself, and once for every parent node of that code in the OMOP ontology up to the root node of our ontology. Consider the ICD10 code E10.1 (Type 1 diabetes mellitus). Any occurrence of this code in a patient's timeline should also give the patient "credit" for having the parent codes of E10.1 – E10 (Type 1 diabetes mellitus) and E08–E13 (Diabetes mellitus). This is because having E10.1 implies that the patient has E10 and E08-E13. We leverage existing OMOP ontology tools for ontology expansion and map codes to their ancestors. Then, when constructing our count matrix $\mathbf{M}$, we count each occurrence of a code for both that code and all of its parent codes. We refer to this ontology-expanded version of our count matrix as $\mathbf{M}'$.

Once the ontology-expanded count matrix $\mathbf{M}'$ is generated, a LightGBM model is trained on this input to predict the target label for each task [17]. Hyperparameter tuning is performed on a validation set following the schedule described in Table 9.

## D.3    CLMBR-T-base

For CLMBR-T-base, each unique medical code $c \in \mathcal{C}$ is associated with a $d$-dimensional embedding $e^c \in \mathbb{R}^d$. Each medical code $x_{pi} = c$ in patient $p$'s timeline is associated with both a code embedding $e^c$ and a position embedding $e^s$ which is defined using rotary positions embeddings [42]. Thus, the input to the model for $x_{pi}$ is given by the concatenation of these vectors, i.e. $e^c \parallel e^s$. For our model, the code embeddings $e^c$ are generated using a standard embedding layer with a vocabulary size of $|\mathcal{C}| = 65,536$. Though there are more unique codes in our dataset, we only keep the top 65,536 codes with the highest contribution to the overall entropy of the dataset – the rest of the codes occurring in our dataset are discarded in order to keep the size of our model's dictionary tractable.

Lab values were discretized by computing decile statistics over the entire dataset and then creating tokens for each lab / decile pair. For example, if the 40th percentile of weight is 180 pounds and the 50th percentile is 190 pounds, we would create one token for "Weight/180-190" which would represent all events with values in that range.

Given this fixed dictionary, a classification task is defined to predict the next code in a patient's timeline given their preceding codes. We use a transformer as our classification model. Our transformer uses a local attention mechanism with a fixed context window of 496 tokens (i.e. clinical events) per layer. As our CLMBR model contains 12 stacked layers, this gives our model an effective context window of 496 * 12 = 5,952 clinical events, on which it conditions to generate its output at each step $i$. Sequences longer than that were truncated.

The output at step $i$ is a $d$-dimensional vector representation of the cumulative information up to and including event $x_{pi}$. We stack these representations for patient $p$ into a matrix $\mathbf{R_P} \in \mathbb{R}^{|\mathbf{X_P}| \times d}$ such that $\mathbf{R_{P_i}}$ is the cumulative $d$-dimensional representation of all events up to and including event $i$ for patient $p$. We then take the dot product of each row in this matrix with every code embedding $e^j$ for all $j \in \mathcal{C}$ in order to calculate a logit for each code $j$ at each event $i$, thus yielding: $\text{logit}_{pij} = \mathbf{R_{P_i}} \cdot e^j$. The model is then trained end-to-end using standard cross-entropy classification log-likelihood loss,

employing an indicator variable $I_{pij}$ to mark if the next event for patient $p$ after event $i$ is an event with code $j$.

The overall loss function, $L(I|\text{logit})$, is computed as:

$$L(I|\text{logit}) = \prod_{p,i,j} I_{pij} \cdot \text{softmax}(\text{logit}_{pi})_j$$

For training our model, we use the best hyperparameters identified in [8] and perform a limited hyperparameter search as defined in 8.

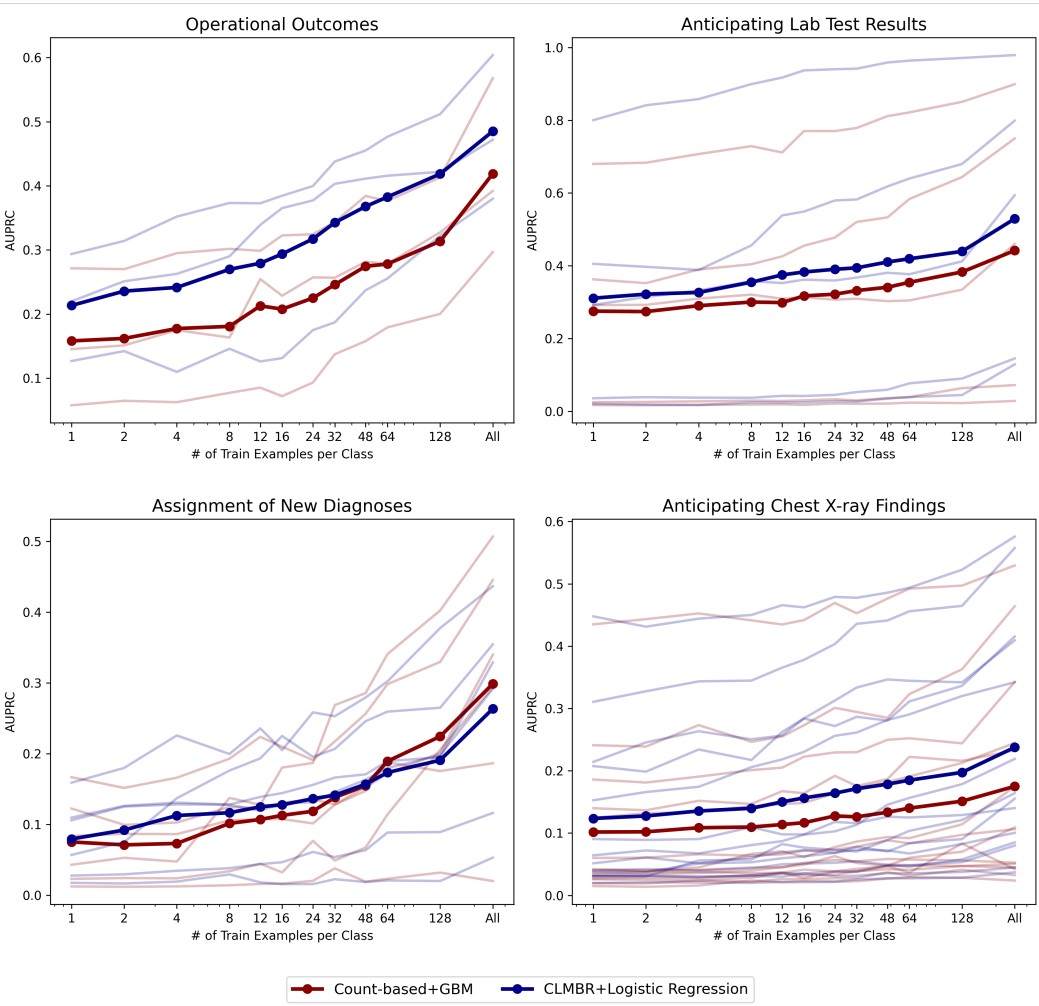

Figure 5: Aggregated AUPRC across all subtasks within each of the 4 task categories for $k \in \{1, 2, 4, 8, 12, 16, 24, 32, 48, 64, 128\}$ shots. We also show performance on the full training set as *All*. The **bolded lines** are the Macro-AUPRC for each model, averaged across all subtasks within a task category for each value of $k$. The blurred lines are the average AUPRC across 5 replicates for each subtask within a task category. Similar to the case with AUROC, the pretrained foundation model CLMBR-T-base (blue) performs better across all $k$ on the *Operational Outcomes*, *Anticipating Lab Test Results*, and *Anticipating Chest X-ray Findings* tasks, while the count-based GBM model (red) performs slightly better at higher $k$ on the *Assignment of New Diagnoses* tasks.

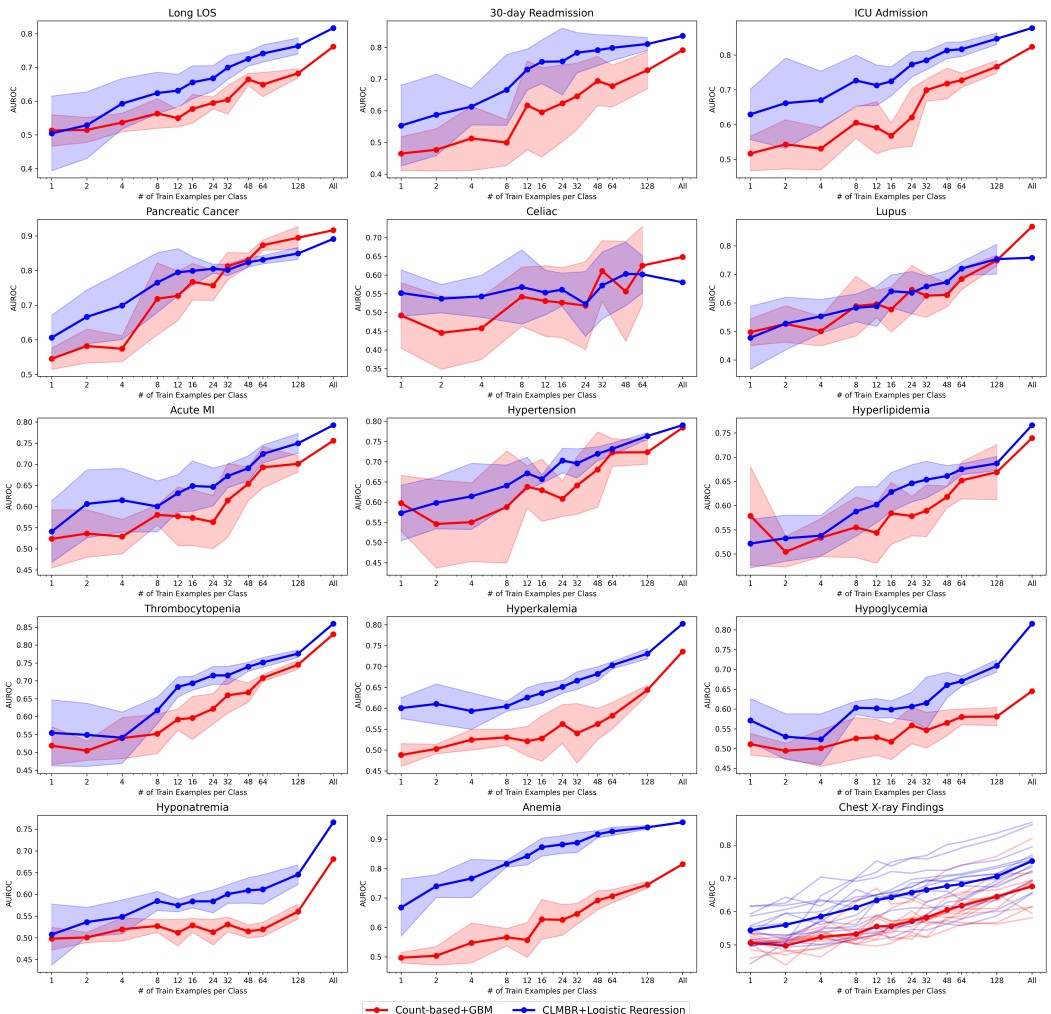

Figure 6: AUROC scores for each model across $k \in \{1, 2, 4, 8, 12, 16, 24, 32, 48, 64, 128\}$ shots. We also show performance on the full training set as *All*. The pretrained foundation model CLMBR-T-base (blue) shows stronger performance on *Operational Outcomes* and *Anticipating Lab Test Results* tasks, while the count-based GBM model (red) exhibits competitive performance at higher values of $k$ for the *Assignment of New Diagnoses* tasks. For *Chest X-ray Findings*, each blurred line represents one of the 14 individual labels, and the bolded line is macro-AUROC across all labels.

Table 8: CLMBR-T-base Hyperparameters

| Name | Values | Best Value |
|------|--------|------------|
| Learning Rate | 0.0001, 0.00001 | 0.00001 |
| Context Window Size | 496 | 496 |
| Internal Dropout | 0, 0.2, 0.4 | 0 |
| # of Layers | 6, 12 | 12 |
| LR Head Learning Rate | 1e-6, 1e-5, ..., 1e5, 1e6 | Task dependent |
| Hidden dimension | 768 | 768 |

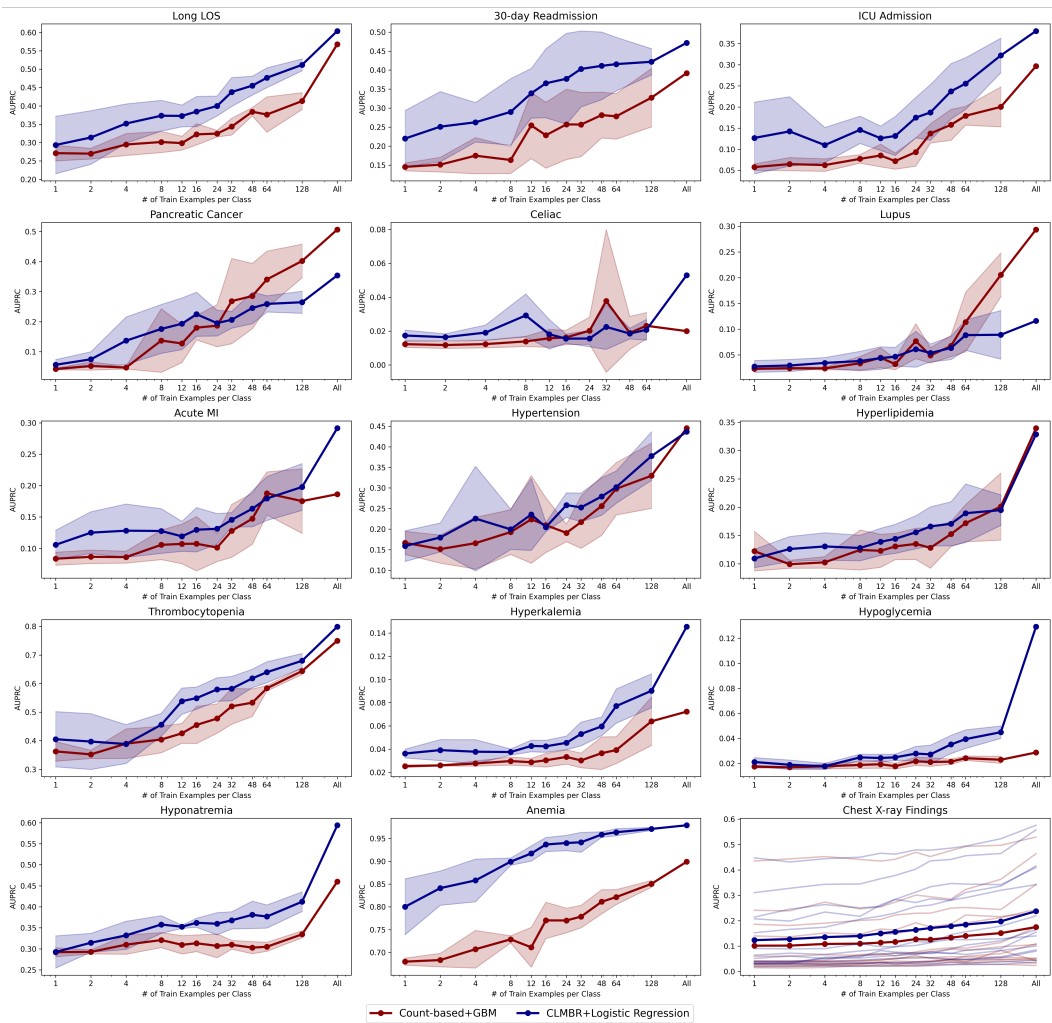

Figure 7: AUPRC scores for each model across $k \in \{1, 2, 4, 8, 12, 16, 24, 32, 48, 64, 128\}$ shots. We also show performance on the full training set as *All*. CLMBR-T-base is in blue, count-based GBM model is in red. For *Chest X-ray Findings*, each blurred line represents one of the 14 individual labels, and the bold line is macro-AUPRC across all labels.

Table 9: GBM Hyperparameters

| Name | Values | Best Value |
|------|--------|-----------|
| Learning Rate | 0.02, 0.1, 0.5 | Task-dependent |
| Max Depth | 3, 6, -1 | Task-dependent |
| Number of Leaves | 10, 25, 100 | Task-dependent |

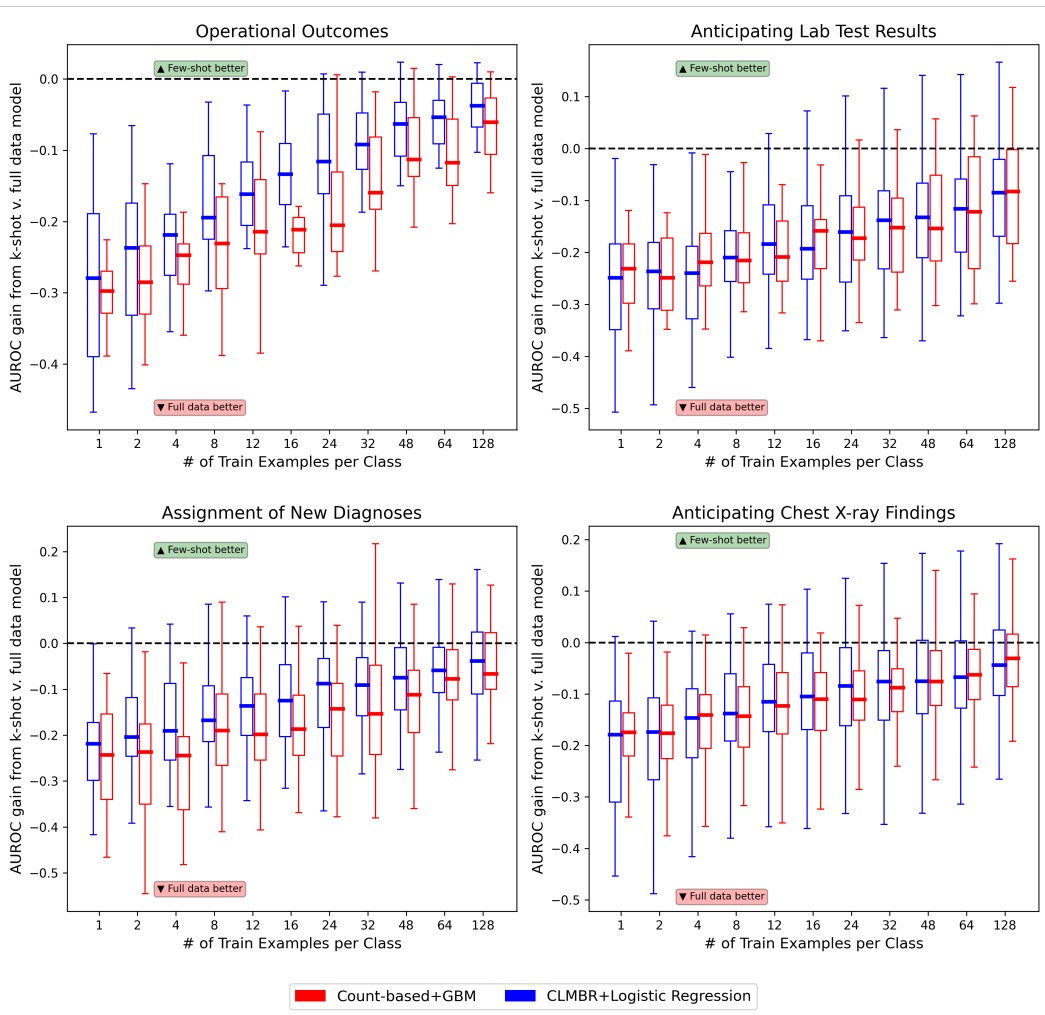

Figure 8: Difference in AUROC between each $k$-shot model replicate and a model trained on the full dataset. The pretrained foundation model CLMBR-T-base (blue) closes the gap with the full data model faster than does the count-based GBM model (red).

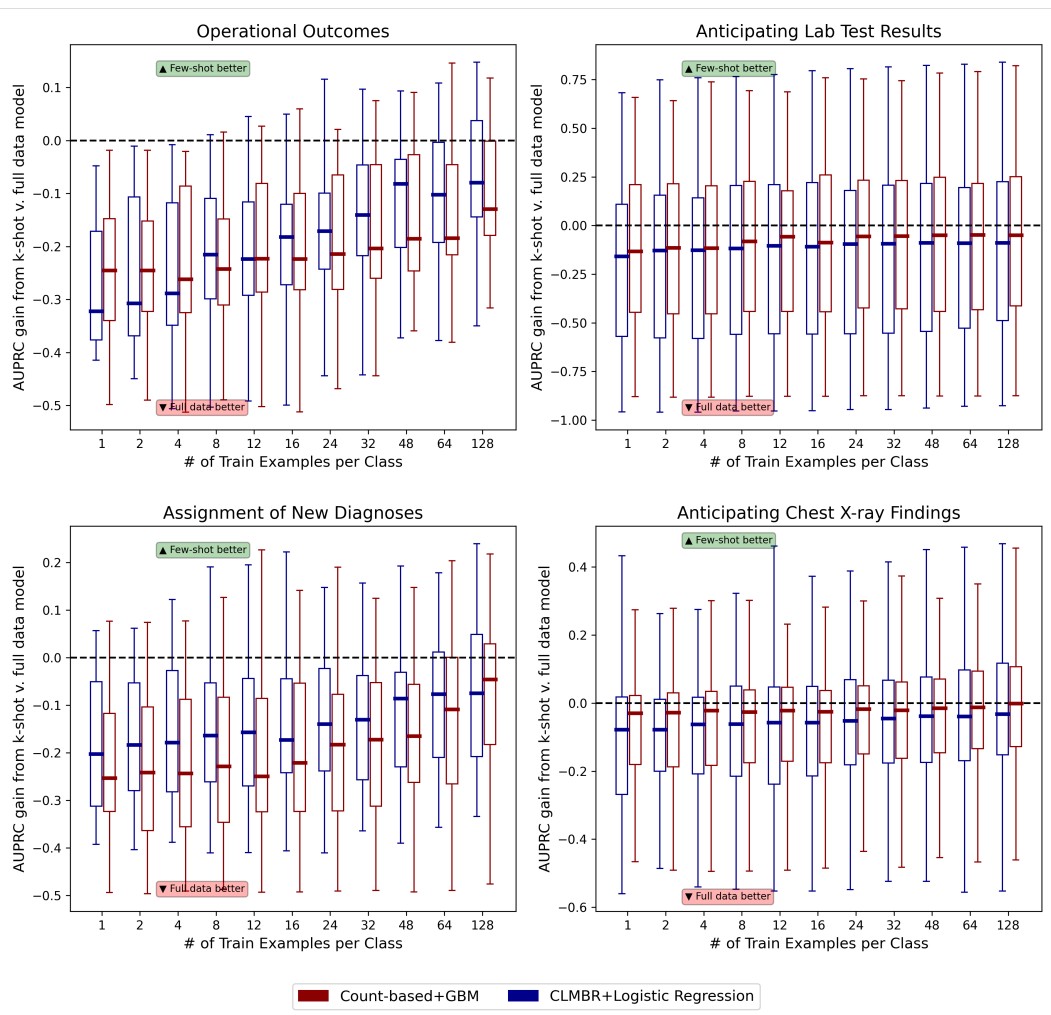

Figure 9: Difference in AURPC between each $k$-shot model replicate and a model trained on the full dataset. The pretrained foundation model CLMBR-T-base (blue) closes the gap with the full data model faster than does the count-based GBM model (red).

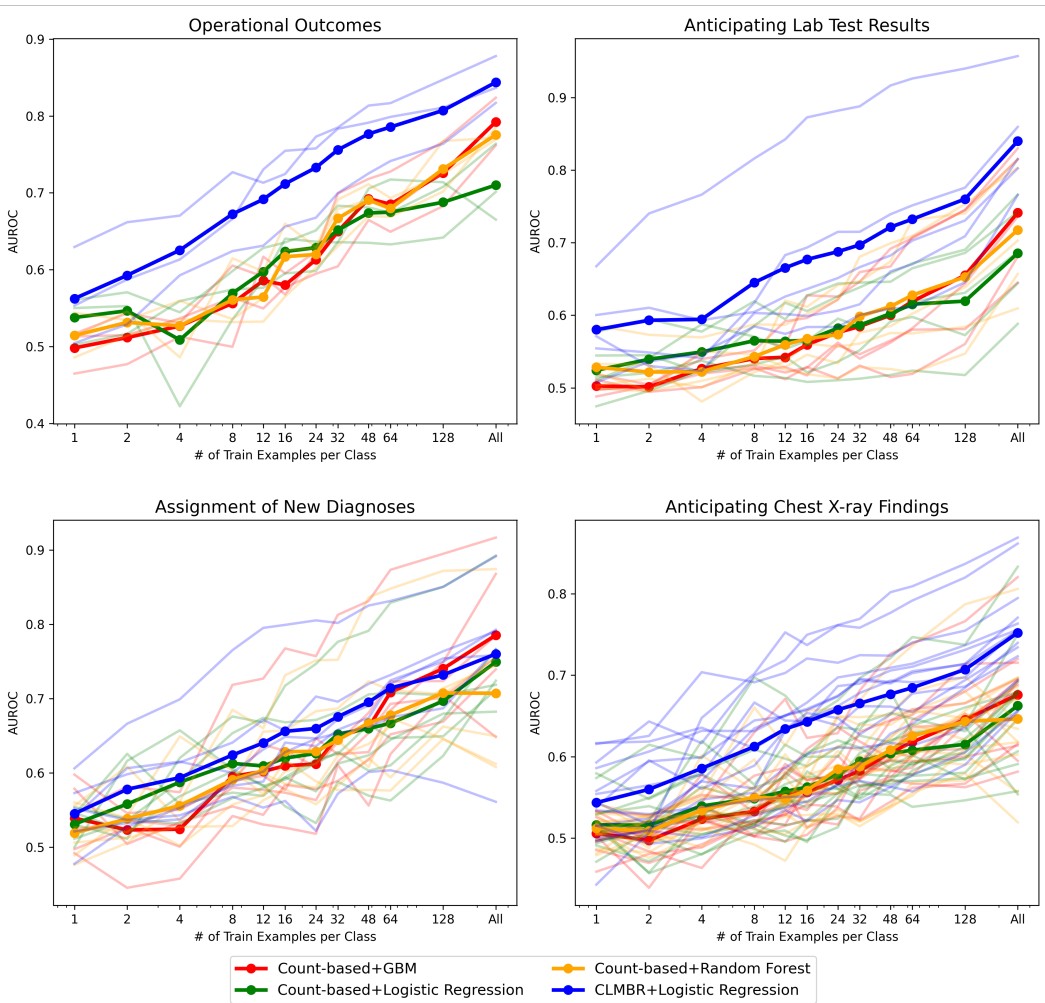

Figure 10: Replication of Figure 3 for aggregated AUROC, but including the following baseline models: CLMBR-T-base (blue), GBM (red), Random Forest (yellow), and Logistic Regression (green).

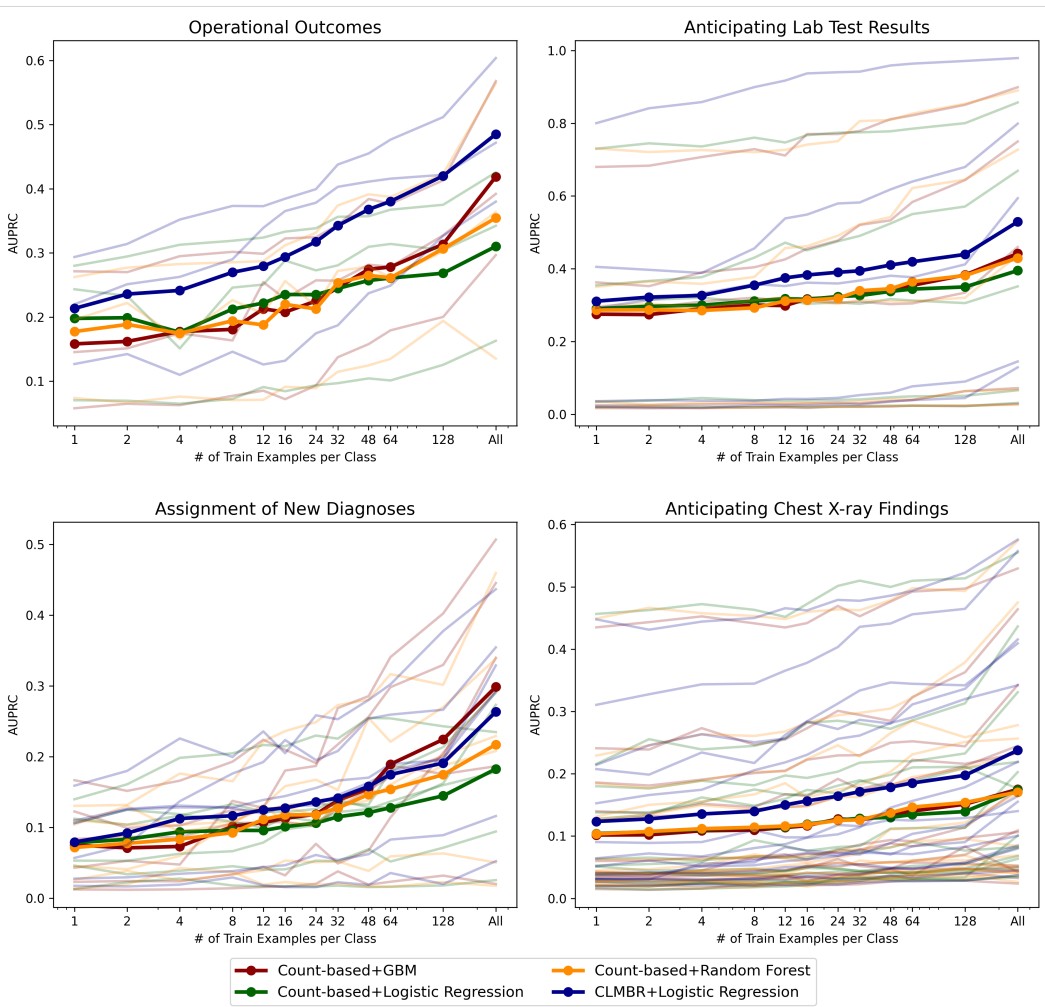

Figure 11: Replication of Figure 5 for aggregated AUPRC, but including the following baseline models: CLMBR-T-base (blue), GBM (red), Random Forest (yellow), and Logistic Regression (green)