# OpenReview forum: "EHRSHOT: An EHR Benchmark for Few-Shot Evaluation of Foundation Models"
_NeurIPS.cc/2023/Track/Datasets_and_Benchmarks — NeurIPS 2023 Datasets and Benchmarks Spotlight_

### Official Review · Reviewer_ipCJ · 2023-07-19
**EHRSHOT is a new EHR dataset for few-shot learning and foundation model evaluation**

**Rating:** 7
**Confidence:** 3

**Strengths:**

-	The authors release the FM weights (hopefully at some point), which will help the broader research community extend the CLMBR model, and also easily benchmark against the method to develop new models.
-	Data is not only from the acute care setting, making it much more realistic than prior datasets and fills a gap in the current literature.


**Additional Feedback:**

Questions

-	What is the size of the embedding dimension for CLMBR?

-	Can you also publish tables as Supplemental Table 3 and 4 but for the pre-training dataset? For a user, it would be helpful to know these high-level statistics to get a sense of how different the populations/distributions might be when fine-tuning the model on a different dataset.
-	Since most lab results are continuous, how were lab results discretized into tokens?
-	In Supplemental Table 4, it shows the mean number of events being at least >5430. What is the maximum sequence length the CLMBR model can take as input, and how are input sequences longer than this maximum sequence length handled (e.g., truncation, random sampling, etc.)?
-	How are the timeline lengths so long? Have some patients really been in the system since birth (e.g., 88 years)? Line 122 says that patient timelines start from 1990.


Comments

-	The CLMBR name is a bit overloaded, given that it’s been used in several other works (but with different underlying architectures) (e.g., Steinberg et al. 2020, Guo et al. 2023). It may be helpful to refer to your proposed new model by a different name, so that future works that cite yours are better able to differentiate between the variations.

Suggestions

-	You can also highlight cost savings for using off-the-shelf CLMBR weights rather than users spending money to pre-train their own model and tune hyperparameters.


**Clarity:**

- Paper is well written.

**Correctness:**

-	The methodology seems sound, though I was unable to access the dataset or FM to verify the construction.

**Documentation:**

-	The documentation is good, and there is a decent amount of commenting to make the code understandable. My only issue is lack of access to the dataset and FM that the paper claims to be releasing.

**Ethics:**

-	Authors have done a nice job of discussing ethical issues and potential societal implications.

**Limitations:**

-	The authors did a nice job of enumerating several important limitations of the work.

**Opportunities For Improvement:**

-	Is GBT a fair comparison as a baseline for few-shot learning? While GBT models are indeed very popular in general practice, I’m not sure they would necessarily be chosen in settings with such large sample sizes.
-	I was unable to download the dataset or model to verify that both are as advertised.


**Relation To Prior Work:**

-	Yes, the authors do a great job making it clear how their FM and dataset fill current gaps in the community given the current literature.

**Summary And Contributions:**

The authors have compiled and released a limited electronic health record (EHR) dataset for fine-tuning and evaluating foundation model (FM) performance in the few-shot learning setting. Additionally, they pre-trained a FM and released the weights for this model and are among the first to do so. They compare the performance of this FM to a baseline model and show improvements in the setting of low sample sizes, which can often in occur in healthcare settings. Overall, both the dataset and the FM are meaningful contributions to the research community and will allow others to further extend the work. However, I was unable to access neither the dataset nor the model, as the Github README claims that they are being reviewed and the download link will be updated once publicly available.

---

> ### Author Response · Authors · 2023-08-23
> **Response by Authors (Part 1)**
>
> Thank you so much for taking the time to write such a detailed review, we really appreciate the suggestions! We are happy to see that we are aligned on the potential benefits of FMs in healthcare, and the need for better evaluation frameworks.
>
> ### Opportunities for Improvement
>
> **GBM as baseline:** The specific type of data included in EHRSHOT is structured EHR data, which is tabular in nature. We observe in the general ML literature that GBT remains a very strong baseline for tabular data compared to deep learning methods [1] [2]. Additionally, specifically within the ML for healthcare literature, gradient boosting methods remain a popular baseline due to their simplicity and strong out-of-the-box performance [3] [4] [5] [6] [7] [8]. Hence why we selected this model as a commonly used and strong baseline to compare against our pretrained foundation model.
>
> [1] Shwartz-Ziv, Ravid, and Amitai Armon. "Tabular data: Deep learning is not all you need." Information Fusion 81 (2022): 84-90.
>
> [2] Gorishniy, Yury, et al. "Revisiting deep learning models for tabular data." Advances in Neural Information Processing Systems 34 (2021): 18932-18943.
>
> [3] Steinberg, Ethan, et al. "Language models are an effective representation learning technique for electronic health record data." Journal of biomedical informatics 113 (2021): 103637.
>
> [4] Tarabanis, Constantine, et al. "Explainable SHAP-XGBoost models for in-hospital mortality after myocardial infarction." Cardiovascular Digital Health Journal (2023).
>
> [5] Chen, David, et al. "Deep learning and alternative learning strategies for retrospective real-world clinical data." NPJ digital medicine 2.1 (2019): 43.
>
> [6] Datta, Suparno, et al. "Predicting hypertension onset from longitudinal electronic health records with deep learning." JAMIA open 5.4 (2022): ooac097.
>
> [7] Desai, Rishi J., et al. "Comparison of machine learning methods with traditional models for use of administrative claims with electronic medical records to predict heart failure outcomes." JAMA network open 3.1 (2020): e1918962-e1918962.
>
> [8] Zhang, Xiaodan, et al. "Comparison of Machine Learning Methods in Mild Cognitive Impairment Prediction for Cancer Patients Using EHR Data." medRxiv (2023): 2023-02.
>
> **Dataset/model access:** Please see our overall comment regarding dataset / model release. But yes, the short answer is there is a plan to release the full dataset/model under a data usage agreement, but given the sensitive nature of EHR data we are still working towards a full model release. We have a clear plan with the stakeholders involved to publish the dataset in a way that meets all necessary requirements and responsible data release guidelines, and the expected date for the full release will be this fall. We will update our Github page as soon as the dataset and model are publicly available.
>
> ### Correctness
>
> **Dataset Access:** Please see above.
>
> ### Documentation
>
> **Dataset Access:** Thank you for the note, and we appreciate your taking the time to read through our code! We will update our Github and README.md accordingly once the dataset is officially released.
>
> ### Comments
>
> **CLMBR Terminology**: Apologies for the confusion! Yes, the model we are using is the same base model that is used in both of those papers, albeit with very slight improvements such as using a transformer instead of an RNN (as in Guo et al. 2023). Thus, it would be accurate to say that "CLMBR" is the same type of model being used by all three papers. Your point that the naming can be much clearer is well taken. Thus, we have renamed our version of CLMBR to be “CLMBR-T-base,” which stands for “CLMBR” implemented with a transformer at a scale similar to BERT-base (141 million parameters, v. 110 million for BERT-base). Additionally, please note that we are the first to actually release the pretrained weights of such a model, and thus hopefully there should not be any confusion for someone looking to download the model weights.
>
> ### Suggestions
>
> **Cost Savings:** Thank you for the comment! That is an excellent point, and one of the primary benefits of using a foundation model. We have added a note to this effect to the Introduction: “Researchers who leverage our model can benefit from both improved downstream task accuracy and cost savings by shortcutting the model development process.”

---

> > ### Author Response · Authors · 2023-08-23
> > **Response by Authors (Part 2)**
> >
> > ### Additional Feedback
> >
> > **CLMBR embed dim:** The size of the embedding dimension for the CLMBR model we trained is 768. We have added a row in our Supplementary Table 6 about CLMBR noting this, and have also added a sentence to our Results section specifying this as well, “Our model has 141M trainable parameters, a hidden dimension of 768, and a next code prediction objective”
> >
> > **Pretraining data statistics:** Thank you for the suggestion. Per your recommendation, we have added two tables to the Supplementary Section of our manuscript (Supplementary Tables 4 and 5) to provide additional detail on the size and statistics of our pretraining dataset.
> >
> > **Lab result discretization:** Lab values were discretized by computing decile statistics over the entire dataset and then creating tokens for each lab / decile pair. For example, if the 40th percentile of weight is 180 pounds and the 50th percentile is 190 pounds, we would create one token for “Weight/180-190” which would represent all events with values in that range. We have added this text to Section D.3 of our Appendix, so thank you for the suggestion!
> >
> > **CLMBR max sequence length:** The maximum sequence length for the CLMBR we used in this paper is 5952 (496 per layer over 12 layers). Sequences longer than that were truncated.
> >
> > **Timeline lengths:** To clarify, by “timeline length” we essentially mean the last recorded age of the patient. We measure the length of each timeline as the time period between the first and last “clinical event” recorded in that timeline. For all patients, the first clinical event will be their birth. This is recorded whether or not a patient was actually born in our source hospital. Thus, while events that occurred at our source hospital started to be recorded in 1990, the “patient birth” event will be backdated to the appropriate time (even though that occurred before 1990 for most patients).

---

### Official Review · Reviewer_gNum · 2023-07-20
**Review for paper 329**

**Rating:** 8
**Confidence:** 5
**Correctness:** The dataset and benchmark are constru…
**Clarity:** The paper is generally well written.

**Strengths:**

The introduction of EHRSHOT fills the gap in the ML for healthcare domain by providing a comprehensive and longitudinal EHR benchmark. This benchmark includes a wide range of clinical events and encounters, making it a valuable resource for evaluating foundation models. The authors contribute to the community by releasing the weights of the CLMBR foundation model. This allows researchers to evaluate and build upon the model, fostering collaboration and advancing the field of ML in healthcare. The paper is generally well-written and easy to follow.

**Additional Feedback:**

Please address my issues above.

**Documentation:**

The authors claim that they will publish the dataset while I didn't found the dataset on the provided website.

**Limitations:**

The authors provide detailed discussion for the limitations.

**Opportunities For Improvement:**

I believe this is generally a good paper. I have a few minor issues:

1. How is longitudinality processed in the model and tasks? Does the model make a prediction at each visit, utilizing all previous visits to learn the embedding (for the CMLBR model)?

2. According to Figure 3, it is impressive that both the GBM and CLMBR models can provide equivalent performance (especially in terms of AUPRC) using just tens of samples. Sometimes, I noticed that the few-shot learning setting even outperforms the full setting. More discussion on this from the authors would be appreciated.

3. I may have overlooked this - It would be beneficial if the authors could provide some quantitative results on how the pretrained CLMBR model can improve the performance on few-shot tasks or other datasets.

4. I checked the dataset webpage but I didn't find the EHRSHOT dataset. Is there a plan to publish this dataset?

**Relation To Prior Work:**

The related works are clear.

**Summary And Contributions:**

This paper presents a benchmark dataset and also a pretrained model aimed at few shot learning healthcare. The authors address the lack of shared assets in ML for healthcare by introducing EHRSHOT, a longitudinal EHR benchmark. This benchmark contains the structured data of 6,712 patients' full medical timelines, including demographics, diagnoses, procedures, laboratory results, medications, and other relevant information. The dataset encompasses a total of 39.2 million clinical events across 893,773 encounters. Additionally, the authors designed 15 patient classification tasks and release the weights of a 141M parameter foundation model pretrained on the deidentified structured data of 2.57M patients' EHRs.

---

> ### Author Response · Authors · 2023-08-23
> **Response by Authors**
>
> Thank you for your time and suggestions for improving this work, we really appreciate the thoughtful comments and are glad to hear that the importance of creating such assets for the ML for healthcare community resonated with you.
>
> ### Opportunities for Improvement
>
> **Handling Longitudinality / CLMBR Predictions:** Thank you for the clarification question, we agree we should have been more specific in our problem formulation in our manuscript. To answer. your question, we handle longitudinality by making predictions at a specified time (which are distinct per task, see Table 7 in the Appendix) using all data that was collected prior to the specified time for that patient. For example, in our Long Length of Stay task, we make predictions at 11:59 pm on the day of admission, so we use all data collected on admission as well as all data from prior visits. The exact way that data is processed depends on the featurization strategy, which varies per model. For the GBM model (which uses count featurization), this is done by counting codes during that prior history and generating count features, as described in more detail in Section D.2 of the Appendix. For the CLMBR-T-base model, the transformer considers the 5,952 most recent clinical events prior to the prediction time to generate a representation for a patient. We only process up to 5,952 events, and not all prior history, because transformers require increased processing time as sequence lengths are increased. The 5,952 is calculated as follows: Our transformer uses a per-layer context window of 496 events; there are 12 layers in our transformer, hence an effective context window of 496 * 12 = 5,952 events. We appreciate the comment, and have added additional text to Section D.3 of the Appendix to clarify this.
>
> **GBM v. CLMBR performance:** Thank you for the question, and yes we agree that we were a bit surprised with some of these results. Since originally submitting, we were able to fix a couple bugs in our pipeline and implement a superior optimizer for our CLMBR-Logistic Regression model based on conjugate gradients. We have updated our figures with these results in the re-uploaded PDF, which I think you will find has addressed most of your concerns. For the AUPRC plots, we agree with the assessment that there are several tasks where GBM and CLMBR look fairly identical -- namely, the New Diagnosis tasks. We provide a few reasons for CLMBR’s underperformance on these specific tasks in the Results section (e.g. its objective is mismatched for long time horizon prediction tasks like these), and we believe that these types of tasks included in our benchmark represent a key opportunity for future work in improving the performance of pre-trained models.
>
> **Quantitative analysis of CLMBR:** While we do not evaluate our CLMBR model on other datasets, as the primary focus of this paper is the release of our benchmark EHRSHOT we prioritize evaluating CLMBR on EHRSHOT (additionally, CLMBR has already been evaluated in other settings [1] [2]). However, please note that we do quantitatively show our CLMBR model’s improved performance over GBM in Figures 3, 5, 6, and 7 across AUROC and AUPRC in our manuscript. In these figures, the x-axis is the number of training examples seen by each model (`k`), and the y-axis is the model performance. The gaps between the lines for GBM and CLMBR across varying values of `k` across the x-axis quantitatively demonstrate how the pretrained CLMBR model can improve over the performance of GBM on few-shot tasks. However, we certainly agree that evaluating our model on other datasets would be interesting future work, which is precisely why we are publishing the weights of our pretrained CLMBR model to enable other researchers to evaluate our model on other benchmarks (something that most pretrained clinical FMs have not done).
>
> [1] Steinberg, Ethan, et al. "Language models are an effective representation learning technique for electronic health record data." Journal of biomedical informatics 113 (2021): 103637.
>
> [2] Guo, Lin Lawrence, et al. "EHR foundation models improve robustness in the presence of temporal distribution shift." Scientific Reports 13.1 (2023): 3767.
>
> **Dataset/model access:** Please see our overall comment regarding the specific dataset / model release timeline. But yes, the short answer is that we have a clear plan with all involved stakeholders at our institution to publish the full dataset in a way that reflects our commitment to patient privacy and responsible data release. We will update our Github page as soon as the dataset becomes available, and we appreciate your understanding considering the nature of health record data we are working to publish.

---

> > ### Comment · Reviewer_gNum · 2023-08-30
> > **Response to author rebuttal**
> >
> > Thank you for the clarifications. I think this will be a good benchmark paper.

---

### Official Review · Reviewer_bA6k · 2023-07-21
**An EHR model and a dataset open for researchers to build upon / reproduce results**

**Rating:** 7
**Confidence:** 4
**Clarity:** The paper is well written

**Strengths:**

The strengths of the above paper include:

1. Comprehensive and Longitudinal Dataset: EHRSHOT provides a comprehensive dataset containing structured data from the complete medical timelines of 6712 patients. This longitudinal approach offers a more holistic view of patient health and medical history, allowing for better evaluation of clinical data foundation models.

2. Few-Shot Evaluation Focus: The paper specifically targets few-shot evaluation, which is crucial in the healthcare domain where data is often limited and obtaining large labeled datasets can be challenging. By focusing on few-shot learning, the paper addresses a practical and relevant problem in healthcare machine learning.

3. Diverse Task Set: The paper defines a set of 15 tasks that cover a range of clinical outcomes, including well-studied outcomes and lesser-explored settings. This diversity ensures that the benchmark evaluates models on a broad spectrum of clinical scenarios, making the results more representative of real-world challenges.

4. Open and Reproducible: The authors go beyond providing the benchmark dataset and release the pretrained model weights and code necessary to replicate their results. This commitment to openness and reproducibility promotes transparency in research and facilitates further advancements in the field

5. Large Pretrained Model: The foundation model pretrained on over 2.57M patient timelines provides a strong baseline for comparison and allows researchers to build upon existing knowledge without starting from scratch. This saves time and computational resources while promoting cumulative progress in healthcare machine learning.

6. Real-World Relevance: The use of real medical data from patients' complete timelines enhances the relevance of the benchmark to practical clinical settings. This increases the potential impact of the research on improving patient care and outcomes.

**Additional Feedback:**

The paper is well written and is a good asset to the clinical researchers. The authors would definitely improve the quality of paper if the questions asked above are addressed.

**Correctness:**

The methodology and results look sound. However, if the authors could provide more clarity / add the following in the paper, it would improve quality of the paper.

1. rationale behind removing clinical text - The authors mentioned they limited the model to only structured data and removed images and clinical text. Why was clinical text removed for the model training ? How is removing clinical text an advantage vs not removing?
2. The authors used k = {1, 2, 4, 8, 16, 32, 64, 128} samples for the training. Can the authors clearly state why they want to use k=1? Or are the authors gradually increasing the samples from 1 to 128. A detailed explanation on the sampling and few shot learning would add more value to the paper
3. example of data with label will be helpful. Since most tasks were binary classification, were they for example in the (30-day Readmission) task, were 635 labels positive and the rest negative ( do not qualify for 30 day readmission)? If yes, were the samples always balanced?
4. In reality, for most clinical tasks, the negative sets contain more sample, will this model perform equally well in such tasks?

**Documentation:**

Yes

**Limitations:**

Yes. The authors very clearly provided limitations in the paper. However, the benefits outweigh limitations in this paper as EHR data is not as easily available as other datasets.

**Opportunities For Improvement:**

The following are the limitations of the paper

1. Dataset Size and Representativeness: While the dataset contains structured data from 6712 patients, it might still be limited in representing the diversity and complexity of clinical cases in real-world healthcare settings. A larger and more diverse dataset would enhance the generalizability of the benchmark.

2. Few-Shot Evaluation : While few-shot evaluation is valuable, it might not fully capture the complexity and performance of models in real-world scenarios. It may not fully represent the challenges faced by models in handling scarce and diverse medical data.



**Relation To Prior Work:**

The authors clearly discussed in table 1 how their work is different from previous contributions

**Summary And Contributions:**

The paper introduces EHRSHOT, a benchmark dataset with structured data from 6712 patients' complete medical timelines, designed for few-shot evaluation of clinical data foundation models. It covers 15 tasks, including 30-day readmission and predicting abnormal lab values. The authors also provide pretrained model weights and code to encourage reproducible and open model development in healthcare machine learning.

---

> ### Author Response · Authors · 2023-08-23
> **Response by Authors (Part 1)**
>
> Thank you for taking the time to provide your comments, and we are glad that so many of the motivating points resonated with you and really appreciate the detailed feedback!
>
> ### Opportunities for Improvement
>
> **Dataset size and representativeness:** Thank you for the note. We definitely agree that including more patients in our dataset would be beneficial, and we certainly aim to add more patients to our benchmark representing a more diverse range of demographics and diagnoses in the future. However, we feel it is important for the research community to engage with this first version of the benchmark, potentially identifying further axes for improvement, before we move to release more patients. We believe that the diverse mix of 15 different clinical tasks that we used to construct our cohort has helped to induce a broad diversity of patient archetypes in our dataset. We do hope to release additional data in future versions of EHRSHOT.
>
> **Few shot evaluation:** First, we would like to clarify that while we focused our analysis on the few shot setting, we do also provide results on evaluating our model on thousands of patients, i.e. our full dataset. This can be seen in the “All” marker at the far right of Figure 3 of our updated manuscript. Second, while we certainly agree that our benchmark does not capture the full complexity of real-world medical scenarios, one of the main challenges of healthcare data is how expensive labels are to obtain. Thus, our focus on the few-shot setting is both (a) representative of how many clinical tasks will be provided to ML models in real-world scenarios, and (b) a unique contribution to the field of ML for healthcare, as most other benchmarks are not focused on few-shot evaluation. Third, while we agree that it is impossible for our dataset to capture the full diversity of medical data, we believe this is a unique strength of our work compared to other benchmarks which are primarily limited to the ICU setting. As our dataset contains full longitudinal data, it provides a uniquely realistic reflection of the scope of data to which a health system would have access. Fourth, the 15 tasks that we define provide much broader coverage than most other EHR benchmarks which are primarily focused on simple-to-define tasks such as mortality / readmissions prediction, and thus our benchmark gives a more complete picture of model performance than other benchmarks. To your overall point, however, we definitely agree that benchmarking should always aim towards being as realistic as possible, and we intend to continue adding more tasks and data to our benchmark in the future.

---

> > ### Author Response · Authors · 2023-08-23
> > **Response by Authors (Part 2)**
> >
> > ### Correctness
> >
> > **Rationale behind removing clinical text:** We agree that releasing clinical text would enrich our dataset. However, including clinical text is not feasible at this time, due to the scale and complexity of implementing de-identification protocols to protect patient privacy for text, which requires manual review to ensure HIPAA-compliance before releasing data for research use. To expedite release of EHRSHOT, we restrict ourselves to structured data. Focusing on this category of models (i.e. foundation models for structured EHR data) is a unique advantage of our benchmark, as there are currently dozens of other text-based benchmarks for biomedical/clinical text, but very few for structured EHR data [1]. Please see this survey [2] for additional background as to what makes a clinical foundation model for structured data distinct from a clinical foundation model for text.
> >
> > [1] McDermott, Matthew, et al. "A comprehensive EHR timeseries pre-training benchmark." Proceedings of the Conference on Health, Inference, and Learning. 2021.
> > [2] Wornow, Michael, et al. "The shaky foundations of large language models and foundation models for electronic health records." npj Digital Medicine 6.1 (2023): 135.
> >
> > **Few shot sampling:** To clarify, `k` in our formulation is the number of examples that the model sees per class. Thus, for `k = 1` for the binary classification task of ICU transfer, the model will see one example of a positive case (i.e. ICU transfer) and one example of a negative case (i.e. no ICU transfer). Our evaluation framework steadily increases the number of samples that our models are trained on (i.e. `k`) from 1 up to 128, increasing by powers of two, and evaluating the model separately at each value of `k`. We thank you for your comment, and agree that we could have been clearer -- we have rewritten the description of our sampling procedure in our Results section as follows: “More precisely, we define "$k$-shot evaluation" of a model $M$ on a specific task $T$ as follows. We train $M$ on $k$ positive examples and $k$ negative examples sampled from $T$'s training split. We then select an additional $k$ positive examples and $k$ negative examples from $T$'s validation split, and use these validation examples to select the best hyperparameters for $M$ for task $T$. Finally, we evaluate the AUROC and AUPRC of the best performing version of $M$ on $T$'s entire held-out test split. For tasks where the total number of unique positive examples is less than $k$, we include all positive examples in our training set, and randomly resample positive examples until the total number of training examples seen by the model is $k$. We consider values of $k \in \{ 1, 2, 4, 8, 12, 16, 24, 32, 48, 64, 128 \}$ for all tasks (with the exception of Celiac, for which we limit $k \le 64$ as there are only 62 positive training labels).”
> >
> > **Example of data with label / 635 positive readmissions:** Yes, for 30-day readmission 635 test set labels were positive and the rest were negative. By negative, we mean that the discharge was not followed by a readmission within 30 days. The classes are not always balanced. Please note that we have slightly modified our cohort from our initial submission, as noted in the Overall Response above. Thus, we have added a sentence clarifying this (with the updated label counts) to the Datasets section: “For example, there are 552 positive labels within the test cohort for the Long Length of Stay task, while there are 2,195 total labels, meaning there are 1,643 negative labels. As there are only 1,238 unique patients in this task's test cohort, some patients have multiple labels assigned to them."
> >
> > **Negative sample imbalance:** Yes, the question about negative sample imbalance is one we’ve thought a lot about, as the majority of clinical tasks will be heavily imbalanced towards negatives. For the purposes of our benchmark, in order to publicly release our dataset we needed to limit the size of our patient cohort to a few thousand patients. We also wanted to enable few-shot learning up to $k = 128$ positive examples. This meant that if we preserved the true prevalence for each task, some tasks with low prevalence (i.e. 1%) would need thousands of additional negative patients in order to preserve the true population prevalence in our dataset. Multiplied across 15 tasks, and the size of our cohort would have ballooned into the 10,000's of patients, which would not have been possible to release. Thus, we had to cap the number of negatives we sampled for each task. We believe this still gives a fair reflection of imbalanced class distributions (though not necessarily as extreme as would occur naturally for some tasks). Additionally, please note that the primary metric that we report, AUROC, is invariant to the balance of positives and negatives. Thus, we would not expect our models’ AUROC scores to change even when evaluated on the natural prevalence of these tasks.

---

### Official Review · Reviewer_G23w · 2023-07-22
**Review of EHRSHOT: An EHR Benchmark for Few-Shot Evaluation of Foundation Models**

**Rating:** 7
**Confidence:** 4
**Correctness:** Yes.
**Clarity:** Yes.

**Strengths:**

- New large-scale structured EHR dataset extends beyond ICU data, unlike previous datasets such as MIMIC-III/IV.
- First to publish the full weights of a clinical foundation model
- Paper is clearly written and easy to follow

**Additional Feedback:**

No.

**Documentation:**

Yes.

**Ethics:**

No.

**Limitations:**

The authors point out that previous works lack the sharing of models and this hinders reproducibility. Have the authors considered releasing the full pretrain data along with the pretrained model? Computer vision and natural langauge processing communities have grown so much because of the open-source pretrained data like ImageNet or C4 datasets. Otherwise this should be considered as one of the limitations of the work.

**Opportunities For Improvement:**

It would be great if the authors can reveal the compute resources used and time spent to pretrain the model

**Relation To Prior Work:**

Yes.

**Summary And Contributions:**

This paper proposes EHRSHOT, a novel EHR dataset that contains patients' full health trajectory during their hospital stay. The authors also release a foundation model pretrained on structured EHR data and define 15 few-shot clinical prediction tasks to evaluate pretrained EHR models.

To construct the dataset, the authors started with Stanford Healthcare EHR data, which consists of 3.67 million patients. The raw data was split into train, valid, and test sets. Using the train set, CLMBR, an EHR pretrained model, was trained, and all the splits were preprocessed and underwent cohort selection, resulting in 6,712 patient in the final dataset. Additionally, the authors defined 15 clinical few-shot classification tasks based on input from clinicians and prior benchmarks, which enabled the evaluation of EHR foundation models for the community.

---

> ### Author Response · Authors · 2023-08-23
> **Response by Authors**
>
> Thank you for your review! We appreciate the time and opportunity to address your questions.
>
> ### Opportunities for Improvement
>
> **Compute Resources Used:** The experiments were performed on an on-premise HIPAA compliant university server with 24 Intel Xeon 2.70GHz CPU cores and 8 Nvidia V100 GPUs. The model itself was pretrained for 4 days on a single Nvidia V100 GPU using 6 CPU cores. We have added a note to this effect to our Results section.
>
> ### Limitations
>
> **Release of Full Pretraining Dataset:** We agree that releasing larger datasets is a worthwhile goal, and that adding additional data to EHRSHOT would be nice to have. However, even creating a research dataset of ~7,000 patients entailed a considerable amount of review and protocols to ensure patient safety. Releasing our full pretraining dataset, which contains over 2.5 million patients, would be infeasible from a governance and effort perspective at this time. We rephrased the following section of our Discussion section to more explicitly mention dataset size as a limitation: “Third, we release a very small cohort of patients (<1\%) from our source EHR database, and specifically select these patients for the tasks that we define. Releasing our full pretraining dataset would be infeasible from a governance and effort perspective at this time. Thus, while necessary in order to publish our EHR dataset and still broader than existing ICU-specific datasets, our cohort selection process limits the types of questions we can answer and does not reflect the full diversity of patients found in hospital EHR system.”

---

### Author Response · Authors · 2023-08-23
**Author Rebuttal by Authors**

We first want to thank all of our reviewers for taking the time to read over our manuscript and write such thoughtful comments and questions. We are glad that the reviewers appreciated the novelty of our work in introducing a longitudinal dataset of EHR data and benchmark, as well as the importance of encouraging transparency and reproducible science for foundation model development for clinical applications. We hope that EHRSHOT contributes to the growing effort to move ML for healthcare towards a culture of reproducible benchmarking and open source models.

We provide a few high-level comments below, and will respond to each reviewer directly as well.

## Summary of our work

Our main contribution is EHRSHOT, a new dataset containing de-identified structured data from the full electronic health records (EHRs) of 6,739 patients. Whereas most prior EHR benchmarks (e.g. MIMIC) are limited to the ICU setting, EHRSHOT is longitudinal and not restricted to ICU/ED patients. Additionally, we provide two unique resources to the community: (1) The weights of a 141M parameter clinical foundation model pretrained on the structured EHR data of 2.57M patients. (2) Fifteen few-shot clinical prediction tasks with a standardized preprocessing pipeline for reproducibility. We release the code publicly on Github, and will provide access to our model / dataset through a Data Usage Agreement via Stanford’s AIMI Center.

## Access to Dataset / Model

Given the sensitive nature of EHR data (hence why there are so few public benchmarks like ours), we are unfortunately unable to release the full dataset and model weights at this moment. In order to best protect patients and our source institution, we have opted to undergo an additional round of safety validation to ensure that the data being released does not pose any risk to patients, providers, or our source hospital. We acknowledge that access to a representative sample of the dataset is critical for the reviewing process, and sincerely apologize about delays in providing a sample to reviewers. While we do have formal approval from our Office of General Counsel, University Privacy Office, University Research IT department, and the hosting site (Stanford AIMI Center) to release this dataset, we encountered some unanticipated delays during the institutional review process due to the breadth of longitudinal data in our dataset. Thus, we are now working with all parties to ensure that the entire dataset will be ready for release as quickly as possible.

In the interim, we have published a representative sample of EHRSHOT at the [Stanford AIMI Center website](https://stanfordaimi.azurewebsites.net/datasets/54c5d2a1-ef39-489e-b5f8-8f9e27e9750b). Once the full dataset is ready for release, it will be hosted at this link as well, and made accessible to researchers who sign a standard Data Use Agreement. The Stanford AIMI Center currently hosts dozens of datasets, and is committed to the long-term archival storage of sensitive data such as EHRs. Out of an abundance of caution, we have also slightly modified our cohort to exclude minors and patients with < 10 clinical events. Our new cohort is now 6,739 patients, rather than the previous 6,712. Updated demographics are available in Tables 2, 3, and 4. We have updated our benchmarking results accordingly, although there has not been any change to the general trends or conclusions.

Our updated release timeline is:

1. October 1st: Finalize license approval
2. November 1st: Complete full dataset + model review
3. December 1st: Release full dataset + model weights before the conference

We do not anticipate any issues, but should this timeline change we will promptly communicate updated timelines to the NeurIPS organizers. We appreciate reviewers’ patience as we work to ensure that the dataset is released in a manner that protects patient privacy  while also fulfilling the needs of the research community. We hope this abbreviated sample allows reviewers to confirm the nature of our dataset.

Though longer than the typical ML dataset release process, we believe this is a positive aspect of our benchmark which reflects our abundance of caution in minimizing risk and ensuring all relevant stakeholders are on board. This will ensure our dataset can be used ethically by researchers going forward as a permanent asset to the ML community. We believe that balancing the joint aims of (a) providing a novel data resource to the ML community for developing models on real-world EHR data and (b) protecting patient privacy and minimizing risk of data leakage justify this slight delay in the official release of our full dataset and model. We will update our Github as soon as both are available

All of our code and analyses are already public on our Github, and will immediately integrate with the full dataset once public.

## Updated Manuscript

Please note that we have submitted a revised manuscript, per the changes discussed in our responses below.

---

### Decision · Program_Chairs · 2023-09-22

**Decision:**

Accept (Spotlight)

**Comment:**

The paper introduces a benchmark dataset with structured data from 6712 patients' electronic health records, designed for few-shot evaluation of clinical data foundation models.
The authors also provide a pre-trained model and code for reproducibility and open model development in healthcare machine learning.
All the reviewers are very positive.